# Atmospheric Chemistry Experiment (ACE) v.5.3 Winds: Validation and Model Comparisons

Matthew Wyatt[1], Peter F. Bernath[1,2,3], Chris Boone[3], Léo Lavy[2], and Ryan Johnson[1]

[1]Department of Physics, Old Dominion University, Norfolk, VA, United States, 23529
[2]Department of Chemistry and Biochemistry, Old Dominion University, Norfolk, VA, United States, 23529
[3]Department of Chemistry, University of Waterloo, Waterloo, ON, Canada, N2L 3G1

**Correspondence:** Matthew Wyatt (mwyat001@odu.edu)

**Abstract.** The Atmospheric Chemistry Experiment Fourier Transform Spectrometer (ACE-FTS) uses limb geometry to measure transmittance spectra of Earth's atmosphere by solar occultation. Line-of-sight wind speeds can be derived via Doppler shifts of molecular lines in infrared spectra. The wind look direction angles relative to geodetic north are derived from geometry. We validate the new ACE version 5.3 (v.5.3) line-of-sight winds with MIGHTI and meteor radar vector wind observations

and find a $-15$ m/s ($+15$ m/s) sunrise (sunset) shift above 80 km. We also compare line-of-sight winds from ACE-FTS v.5.2 and v.5.3 with vector winds from the MERRA-2, HWM14, and WACCM-X models. A $-15$ to $-20$ m/s ($+15$ to $+20$ m/s) sunrise (sunset) bias persists in v.5.3 winds above 80 km but decreases to less than $-5$ m/s ($+5$ m/s) below 50 km. The v.5.3 wind speed profiles have improved relative to v.5.2 at all altitudes. Over 20 years of ACE wind speeds can be used to test atmospheric models.

*Copyright statement.* TEXT

## 1 Introduction

The Atmospheric Chemistry Experiment (ACE) mission on board the Canadian satellite SCISAT is used for remote sensing Earth's atmosphere (Bernath et al., 2005; Bernath, 2017). The SCISAT satellite operates in a low Earth near polar orbit with an inclination of $73.9°$. The ACE mission uses limb geometry during solar occultations to measure the transmission spectra of the

atmosphere. Using the Sun as a light source, the high-resolution Fourier transform spectrometer (ACE-FTS) on board collects spectra as the Sun rises and sets from the perspective of the satellite. Line-of-sight wind speeds can be derived from the Doppler shift of spectral lines in the limb measurements. This is possible due to the frequency stability of the Michelson interferometer. The major advantages of ACE winds are the large altitude range, 18 to 140 km, and the longevity of the mission, starting in 2004 and still ongoing.

ACE wind speed retrievals have been described in detail for v.5.2 of ACE-FTS processing (Boone et al., 2021). A validation of v.5.2 line-of-sight winds has been carried out by comparing against meteor radar and remote sensing data (Johnson et al.,

2024). Version 5.3 processing has been released for ACE-FTS data products, which includes an update to the retrieval of line-of-sight wind speeds. A detailed explanation of the new wind retrieval process is included in this work.

Winds measurements by ACE are taken from the lower stratosphere through the lower thermosphere; however, most instruments only cover a small altitude range in comparison. Horizontal wind speeds are available in the troposphere and lower stratosphere through measurements from airplanes (Khelif et al., 1999) and balloons (Duruisseau et al., 2017; Kumer et al., 2014), ground-based lidar (Martner et al., 1993), and satellite lidar from the ADM-Aeolus (Atmospheric Dynamics Mission Aeolus) (Stoffelen et al., 2005; Lux et al., 2020). Ground-based lidar measurements have also been successful in the middle atmosphere (Baumgarten, 2010; Liu et al., 2002). Vector wind measurements in the upper mesosphere lower thermosphere (UMLT) can be recorded from space using Doppler shifts in airglow lines such as from atomic oxygen. A current example of this is the TIMED Doppler Interferometer (TIDI) instrument on the Thermosphere Ionosphere Mesosphere Energetics and Dynamics (TIMED) satellite (Killeen et al., 2006). Previously, the Michelson Interferometer for Global-High-resolution Thermospheric Imaging (MIGHTI) instrument on the Ionospheric Connection Explorer (ICON) satellite used a similar technique (Englert et al., 2017). On the Upper Atmosphere Research Satellite (UARS), the High Resolution Doppler Imager (HRDI) used $O_2$ emission to measure winds in the UMLT (Hays et al., 1994; Grassl et al., 1995). Ground-based meteor radar (Liu et al., 2002; Tang et al., 2021) can also provide winds in the mesosphere. Line-of-sight winds near the mesopause have also been derived from the Doppler shift in $O_2$ emission lines by the Microwave Limb Sounder (MLS) instrument on the Aura satellite (Wu et al., 2008). Since many of the missions mentioned are inactive, only measure a portion of the altitudes ACE covers, or only cover a small portion of the globe, the line-of-sight winds from ACE are especially valuable.

In this work, three wind datasets are directly compared to the new v.5.3 of ACE winds. The Modern-Era Retrospective Analysis for Research and Applications, version 2 (MERRA-2) produced by NASA's Global Modeling and Assimilation Office (GMAO) is an atmospheric reanalysis model based on modern satellite observations. MERRA-2 provides various data collections that contain information about many climate indicators, including atmospheric wind speeds. An in-depth explanation of the model is available from Gelaro et al. (2017). MERRA-2 uses the Goddard Earth Observing System (GEOS) atmospheric model (Rienecker et al., 2008; Molod et al., 2015) and the Gridpoint Statistical Interpolation (GSI) analysis scheme (Kleist et al., 2009). The model reaches to near 75 km, overlapping with the lower half of ACE data altitudes.

Horizontal Wind Model Version: 2014 (HWM14) is an empirical climatology model of horizontal winds ranging from the troposphere up through the thermosphere. A detailed description of the climatology is available from Drob et al. (2015). HWM14 uses $\sim 73 \times 10^6$ observation measurements from 44 different instruments and a set of spherical harmonics to provide a statistical view of vector winds ranging from near the surface up to 1000 km. Because HWM14 is an empirical climatology, vector winds can be found for any given latitude and longitude.

The Whole Atmosphere Community Climate Model with thermosphere and ionosphere eXtension (WACCM-X) is produced by the National Center for Atmospheric Research (NCAR). WACCM-X is a comprehensive numerical model of the whole atmosphere, ranging from Earth's surface up to around 700 km (Liu et al., 2018). The model is an altitude extension of WACCM6 (Gettelman et al., 2019), which reaches up to $\sim 140$ km. WACCM outputs many climate and weather data products and is unique in that the model can be coupled with others to include ocean, sea ice, and land components. Using CESM2

(Community Earth System Model 2) as a framework, WACCM is a mesh of NCAR projects: High Altitude Observatory (HAO) in the upper atmosphere, Atmospheric Chemistry Observations & Modeling (ACOM) in the middle atmosphere, and Climate & Global Dynamics (CGD) in the lower atmosphere.

In Section 2 of this paper we provide a brief explanation of ACE v.5.2 line-of-sight wind speed processing and a detailed explanation of v.5.3. Following, in Section 3 we validate ACE v.5.3 line-of-sight wind speeds with MIGHTI and meteor radar data. We then compare both v.5.2 and v.5.3 to the MERRA-2, HWM14, and WACCM-X models in Section 4. In the final section we provide concluding remarks.

## 2    ACE-FTS Line-of-Sight Wind Retrievals

### 2.1    ACE v.5.2 Winds

The retrieval process for ACE-FTS v.5.2 wind profiles has previously been explained in-depth by Boone et al. (2021), so only a short explanation is provided here. For each limb measurement, the atmosphere is partitioned into 4 km regions above 45 km. Below that, increasingly smaller regions are used that decrease to 2 km in size. This is done to account for refraction distorting the disk of the Sun during a measurement. A forward model calculation is then used to derive a spectrum at a tangent height near the center of each partition. This forward model is based on pressure, temperature, and volume mixing ratio (VMR) profiles taken from sr10063 (sr for sunrise and 10063 for the orbit number since launch). The observed spectra and sr10063 are cross correlated to derive wind speeds from the Doppler shift in the spectra. The wind speeds are then interpolated onto a 1 km grid using a cubic spline. Finally, the wind profiles are calibrated using line-of-sight winds from the Canadian Meteorological Center (CMC) between 19 and 24 km altitude. The final product is line-of-sight wind speeds between 18 and 140 km at a given time, location, and heading. The heading is the angle between the 'look-direction' of the instrument and geodetic north, which is calculated by the Systems Tool Kit software package (STK).

    The models and instruments we compare with provide vector wind speeds, not line-of-sight wind speeds. We are able to convert the vector winds into line-of-sight winds using

$$\text{Line-of-Sight Wind Speed} = U \cdot \cos(\theta - 90°) + V \cdot \cos(\theta), \tag{1}$$

where $U$ and $V$ are the zonal (west-east) and meridional (south-north) components, respectively, from the given model or instrument, and $\theta$ is the heading of the ACE-FTS instrument.

### 2.2    ACE v.5.3 Winds

Version 5.3 explicitly retrieves altitude profiles for line-of-sight wind speed simultaneously with retrieving the VMR profiles of molecules contributing to the spectrum. The retrievals follow the usual procedure described previously for VMRs (Boone et al., 2023), but with a piecewise linear interpolation for wind speed onto the 1-km altitude grid employed for forward model calculations. Deriving wind speed using a retrieval, rather than using the wavenumber offset of a given measurement as an estimate of the Doppler shift at that measurement's tangent height (as was done in previous processing versions), should better

capture the altitude variation of wind speed when there is significant structure in the profile. With the retrieval approach, the calculated spectrum employed in the analysis is also more representative of the measured spectrum compared to the "generic"

calculation from a single occultation (sr10063) in a given altitude "bin" that was employed in the v.5.2 analysis.

The altitude range was divided into 5 overlapping segments. The microwindows for a given altitude segment targeted single bands or sets of closely spaced bands (including overlapping hot bands) for a particular isotopologue to maximize internal consistency. It was found that observed shifts for lines from different isotopologues or lines from molecules without distinct bands (like $H_2O$) were not always consistent. The microwindows employed in the different altitude segments are provided in

Table A1 of the Appendix. All measurements that fall within the wavenumber, altitude, and minimum transmittance ranges are employed in the retrieval. The filter for minimum transmittance is used to avoid lines that are saturated or where the signal is too close to the noise, either of which could obscure Doppler shift information in the measurement. The upper altitude limit of a particular segment is determined by the maximum tangent height within the altitude range that has at least three lines satisfying the filter conditions, to avoid having an abnormally large noise feature being included in the analysis when the signal

falls below the nominal noise level.

Wind retrieval results for the lowest altitude segment (termed "segment 1" in Table A1) exhibit excessively large variability, possibly a consequence of pressure shifts and deviations of line shapes from the assumed Voigt profile becoming more prominent at higher pressures. Therefore, in this segment line-of-sight wind speed is determined using the average wavenumber shifts of the lines, similar to the approach employed in previous processing versions, rather than using the retrieved results.

Attempts to push the analysis lower in altitude often yielded relatively large gradients below 18 km, which are not expected in that altitude region, suggesting that systematic errors from pressure-related issues (such as pressure shifts and line shapes) become excessive when pressure gets too high. Therefore, the lower altitude limit of the analysis remained at 18 km.

Wind profiles in the lowest two altitude segments have persistent biases compared to the results in the three highest segments, as shown in Fig. 1. Thus, fixed shifts are applied to the wind profiles in the lowest two segments: -35 m/s in segment 1 and

+35 m/s in segment 2. In overlap regions between segments, results are calculated from an unweighted average of the two profiles. When no measurement occurs in the overlap region between segments, there is a small altitude gap in the reported wind profile. As with previous processing versions, the final wind profile is shifted such that the results between 18 and 24 km match the expectations from an analysis run of the Canadian weather model at Environment and Climate Change Canada (Buehner et al., 2015). This is needed to account for the motion of the satellite relative to the atmosphere. A rough calibration

of the ACE-FTS wavenumber scale is accomplished using high altitude $CO_2$ lines, but because this process uses sampled peaks (which may not be sampled at line centers), the calibration could be off by a fraction of the width of the instrumental line shape (0.02 cm$^{-1}$). The accuracy of this calibration will vary from occultation to occultation, but the wind calibration shifts can be as high as a few hundred m/s to compensate for the resulting offsets.

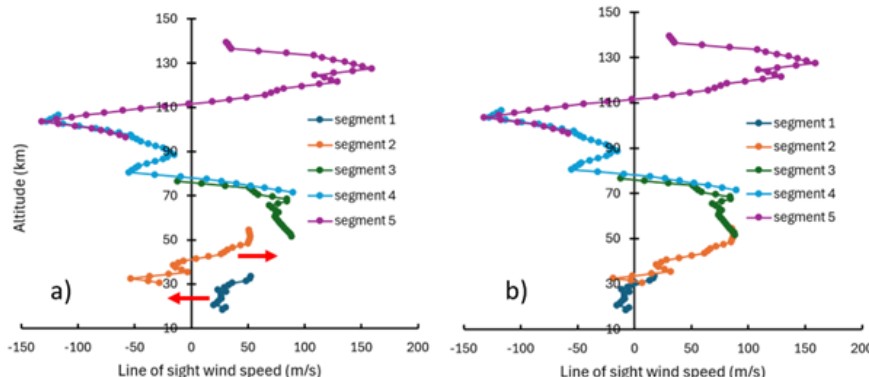

**Figure 1.** Wind retrieval for sr10063 (measured June 25th, 2005, at latitude 37.6° S and longitude 161.4° E). **a)** The "raw" retrieval, with large steps for the profiles in the bottom two altitude segments. **b)** Internal consistency is improved after shifting the profiles in the bottom two altitude segments

### 2.2.1 Calculating Look Direction

Comparing ACE-FTS winds to vector (zonal and meridional) wind information from other sources requires knowledge of the look direction of the instrument relative to either a northward-pointing vector (the positive direction for meridional winds) or an eastward-pointing vector (the positive direction for zonal winds). Previous processing versions employed the Systems Tool Kit (STK) software package to generate "heading" information (the angle relative to a north-pointing vector) associated with the vector extending from the measurement tangent point to the satellite (Boone et al., 2021). However, lacking an STK built-in

function for this purpose, the calculation script represented a kludge of the software that appears to have had accuracy issues, particularly when the measurement tangent point approached the poles.

An alternative approach for the heading calculation is available, based on the fact that ACE ephemerides (satellite position and velocity information) in the Earth-centered Earth-fixed J2000 coordinate system are available on time scales of one minute or less for the duration of the ACE mission. Geodetic information (latitude, longitude, and altitude) for ACE-FTS tangent

points, expressed relative to the WGS84 reference frame, can be readily converted into J2000 coordinates (Zhu, 1994). With satellite and tangent point locations available in the same cartesian coordinate system, heading information can be directly calculated from first principles.

Given the geodetic coordinate of a tangent point in the WGS84 reference frame $(\phi, \lambda, h)$, where $\phi$ is latitude, $\lambda$ is longitude, and $h$ is altitude, the corresponding coordinate of the point in the J2000 cartesian frame, $(x_t, y_t, z_t)$, can be calculated using

the following algorithm (Zhu, 1994):

$$x_t = (N + h)\cos\phi\cos\lambda, \tag{2}$$

$$y_t = (N + h)\cos\phi\sin\lambda, \tag{3}$$

$$z_t = \left((1 - e^2)N + h\right)\sin\phi, \tag{4}$$

where $e$ is the eccentricity of the ellipsoid describing Earth's sea level ($e^2$ is 0.00669437999014 for WGS84), and the radius of curvature ($N$) at latitude $\phi$ is calculated as follows:

$$N = \frac{a}{\sqrt{1 - e^2 \sin^2 \phi}}, \tag{5}$$

where $a$ is the radius at the equator, 6378.137 km in WGS84.

The "look vector" is then simply the difference between the two points

$$(x_t - x_s)\hat{i} + (y_t - y_s)\hat{j} + (z_t - z_s)\hat{k}, \tag{6}$$

where $(x_s, y_s, z_s)$ is the coordinate of the satellite in J2000, and $\hat{i}$, $\hat{j}$, and $\hat{k}$ are unit vectors in the $x$, $y$, and $z$ directions, respectively. However, determining the angle relative to vector winds requires calculating the projection of this vector onto a plane normal to the tangent point that contains either an eastward-pointing or northward-pointing vector. Due to the symmetry of the assumed ellipsoid shape, it is more convenient to find a plane normal to the tangent point that contains an eastward-pointing vector. The equation for the normal plane can be expressed as follows:

$$normal(x, y, z) = \frac{\partial f}{\partial x}(x_t, y_t, z_t) \cdot [x - x_t] + \frac{\partial f}{\partial y}(x_t, y_t, z_t) \cdot [y - y_t] + \frac{\partial f}{\partial z}(x_t, y_t, z_t) \cdot [z - z_t], \tag{7}$$

where, because at a given $z$ the "surface" of interest consists of a circle in the $x/y$ plane:

$$f(x, y, z) = x^2 + y^2 + g(z). \tag{8}$$

The derivative of the function $g(z)$ effectively represents the tilt of the plane as a function of $z$. However, the angle between the projection of the look vector on the plane and a vector in the plane at constant $z$ (which will be parallel to the east/west direction vector at the tangent point) is not sensitive to the tilt of the plane in the $z$ direction. Therefore, for simplicity, $g(z)$ can be set to a constant, chosen to be zero. This makes $f(x, y, z)$ a cylinder with its axis along the z-axis in J2000.

A vector orthogonal to the plane is therefore:

$$\frac{\partial f}{\partial x}(x_t, y_t, z_t)\hat{i} + \frac{\partial f}{\partial y}(x_t, y_t, z_t)\hat{j} + \frac{\partial f}{\partial z}(x_t, y_t, z_t)\hat{k} = 2x_t\hat{i} + 2y_t\hat{j} + 0\hat{k}. \tag{9}$$

The projection of the satellite's position on the plane $(x_p, y_p, z_p)$ can be found using the dot product of the look vector with the (normalized) vector orthogonal to the plane:

$$\beta = \frac{\left((x_t - x_s)\hat{i} + (y_t - y_s)\hat{j} + (z_t - z_s)\hat{k}\right) \cdot (2x_t\hat{i} + 2y_t\hat{j})}{(2x_t\hat{i} + 2y_t\hat{j}) \cdot (2x_t\hat{i} + 2y_t\hat{j})} = \frac{2x_t(x_t - x_s) + 2y_t(y_t - y_s)}{4x_t^2 + 4y_t^2}, \tag{10}$$

$$x_p = x_s + \frac{\partial f}{\partial x}(x_t, y_t, z_t)\beta = x_s + 2\beta x_t, \tag{11}$$

$$y_p = x_s + \frac{\partial f}{\partial y}(x_t, y_t, z_t)\beta = y_s + 2\beta y_t, \tag{12}$$

$$z_p = x_s + \frac{\partial f}{\partial z}(x_t, y_t, z_t)\beta = z_s. \tag{13}$$

The angle ($\theta$) between the projected vector and an eastward-pointing vector in the plane then becomes

$$\theta = \tan^{-1}\left(\frac{z_p - z_t}{\sqrt{(x_p - x_t)^2 + (y_p - y_t)^2 + (z_p - z_t)^2}}\right). \tag{14}$$

Unfortunately, there is an ambiguity regarding whether the vector is directed to the east or to the west. To determine this, the longitude of the satellite ($\phi_s$) is calculated:

$$\phi_s = 2\tan^{-1}\left(\frac{\sqrt{x_s^2 + y_s^2} - x_s}{y_s}\right). \tag{15}$$

By simple geometry, if this longitude is west of the longitude of the measurement tangent point, the projected vector is directed westward from the tangent point, and if it is east of the tangent point, the projected vector is eastward. Knowing the angle of the projected vector relative to an eastward-pointing vector, the heading angle reported in the ACE-FTS wind data (relative to a northward-pointing vector) can be readily determined.

Figure 2 compares the heading angle (relative to a northward pointing vector) determined using the new approach to values
determined previously using STK for sunset occultations in 2019. Both approaches exhibit occasional apparent jumps in heading when the east/west pointing direction can be difficult to decipher, such as when the satellite is on the opposite side of the pole from the tangent point. Overall, there is a general agreement in how the heading angle varies over the course of the year, but differences between the two angles often exceed $20°$. Agreement is best when the heading is close to either north/south or east/west and worst when the tangent point is near one of the poles. The STK calculations exhibit significantly
greater variability.

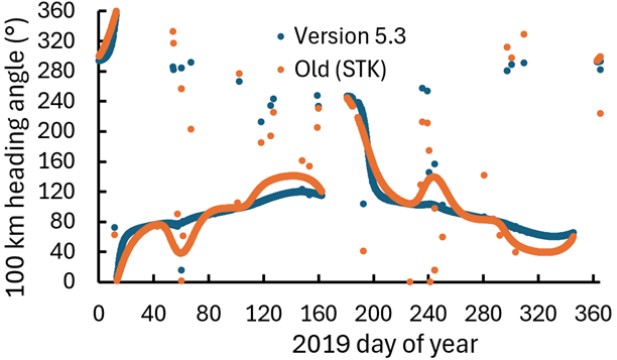

**Figure 2.** Calculated heading angle of the ACE-FTS line-of-sight vector for sunset occultations in 2019. Calculations for v.5.3 winds using J2000 coordinates for both the satellite and the tangent point are shown in blue, while calculations from previous ACE-FTS wind versions using STK are shown in orange.

# 3   ACE v.5.3 Validation

## 3.1   MIGHTI vs ACE v.5.3

The MIGHTI instrument uses a Michelson interferometer and utilizes the Doppler Asymmetric Spatial Heterodyne (DASH) technique to determine shifts in emission lines (Englert et al., 2017, 2023). The emission lines observed are the red ($O(^1D-^3P)$ 630.0 nm) and green ($O(^1S-^1D)$ 557.7 nm) lines of the oxygen atom. We only compare with winds derived from the green line data products, as the red line is less frequent below 150 km. MIGHTI uses two sensors that record measurements 5 to 8 minutes apart. These measurements can be used to determine horizontal vector winds due to being orthogonal to each other, with MIGHTI-A observing at $45°$ and MIGHTI-B at $135°$ relative to the satellite's motion.

A detailed comparison with ACE v.5.2 line-of-sight winds with MIGHTI vector winds is available in Johnson et al. (2024). We provide a similar comparison of ACE v.5.3 against v05 MIGHTI using 207 of the same sunrise and 215 sunset coincidences between 2019 and 2022 used in the previous paper. Those coincidences are measurements within $2.5°$ latitude and $5°$ longitude in location and 2 hours in time. In the case that there were multiple MIGHTI measurements that fit this criteria for one ACE occultation, the spatially closest measurement was used.

In Fig. 3, we show a comparison between an individual ACE sunrise and sunset occultation and the coincident MIGHTI measurement. It is important to note that the heading angle differs in v.5.2 and v.5.3, meaning that the converted MIGHTI winds would be different for each. However, the resulting speeds are very similar, so we display the line-of-sight winds converted using v.5.3 heading angles. There is similar agreement for both v.5.2 and v.5.3 with MIGHTI when looking at the sunrise coincidence. The sunset coincidence shows slightly better agreement using v.5.3.

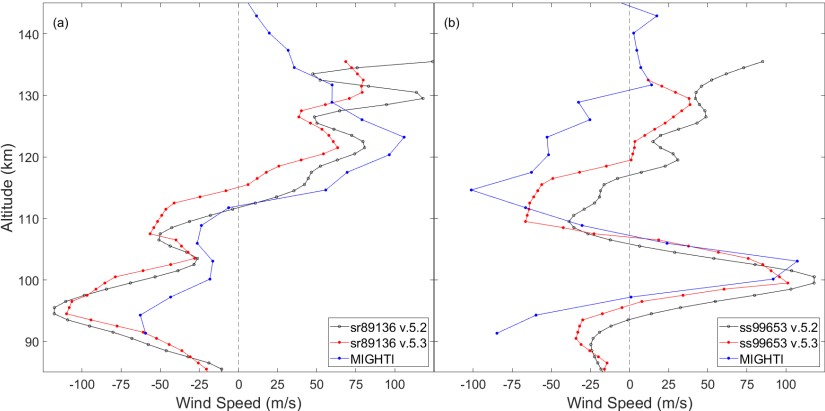

**Figure 3.** Comparison between individual ACE v.5.2 and v.5.3 **(a)** sunrise and **(b)** sunset coincidences with MIGHTI measurements. MIGHTI wind speeds were converted using the ACE v.5.3 heading angle.

In an effort to remove outliers and find trends in the data, we use the same restrictions as Data Set 2 from Johnson et al. (2024). Data Set 2 used only the coincidences where the average difference between ACE and MIGHTI wind speeds was less

than 60 m/s, leaving us with 184 sunrises and 196 sunsets. The outliers removed are typically associated with a limited altitude window for the MIGHTI measurement. The average of these MIGHTI and ACE measurements are presented in Fig. 4. Looking at the sunrises, there is similar profile agreement at all altitudes for both v.5.2 and v.5.3. The sunset comparison shows good agreement for both versions from 90 to 110 km and then deviates.

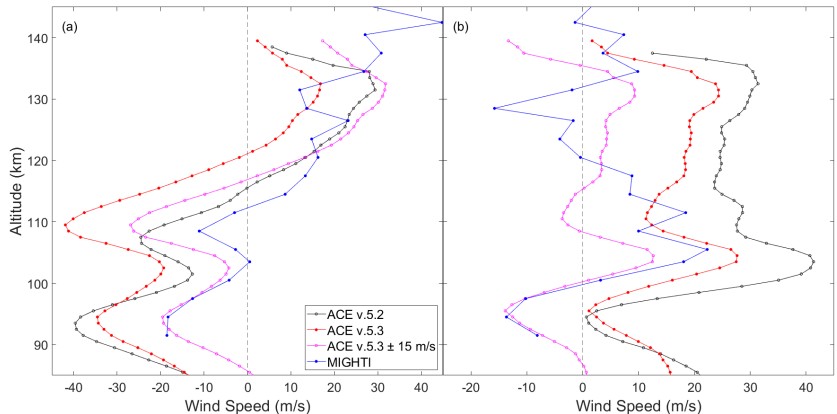

**Figure 4.** Average wind speeds of **(a)** 184 sunrise and **(b)** 196 sunset ACE v.5.2 and v.5.3 coincidences with MIGHTI measurements. MIGHTI wind speeds were converted using the ACE v.5.3 heading angles.

We are also able to derive a sunrise-sunset bias with this comparison. To do this, we shift the MIGHTI altitude to the nearest ACE altitude (maximum of $\pm 0.5$ km) and find the difference between ACE and MIGHTI wind speeds. The differences are then averaged at each altitude over all occultations in Data Set 2 and the results are shown in Fig. 5(a) for v.5.2 and (c) for v.5.3. We then subtract the sunrise and sunset averages from the total average to find the bias. The average bias at each altitude is displayed in Fig. 5(b) for v.5.2 and (d) for v.5.3. We find that the sunrise (sunset) bias for v.5.2 is generally within $\pm 5$ m/s

of the previously found $-15$ m/s ($+15$ m/s) up to 120 km. The bias for v.5.3 is within $\pm 2$ m/s of the $-15$ m/s ($+15$ m/s) bias up through the same altitude. The sunrise-sunset bias is shown in pink in Fig. 4.

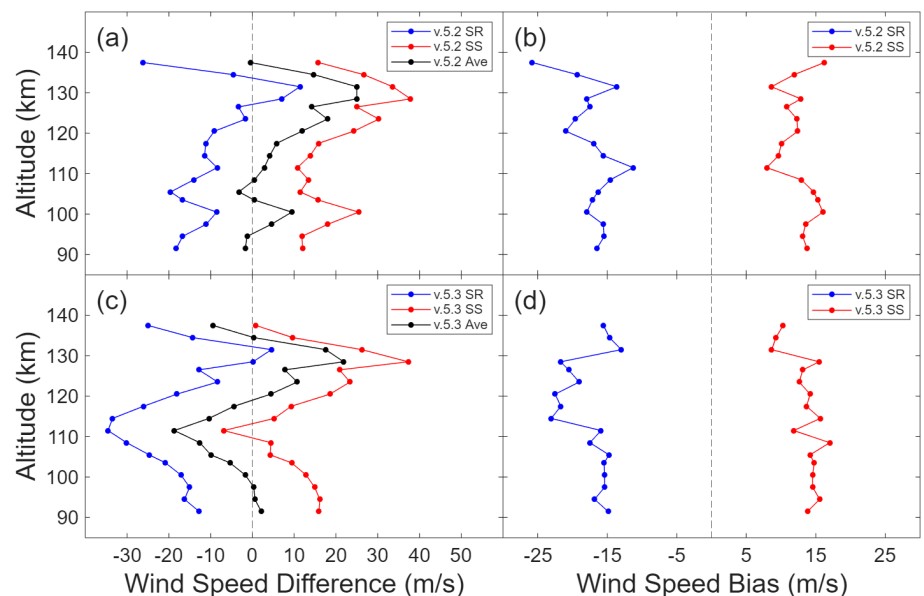

**Figure 5. (a)** Average difference between ACE v.5.2 (sunrise (blue), sunset (red), and both (black)) and MIGHTI winds for each ACE wind speed measurement in 2019. **(b)** ACE sunrise-sunset bias, shown as the difference between sunrise and sunset averages and the total average in (a). Plots **(c)** and **(d)** are the same as (a) and (b), respectively, but for ACE v.5.3 processing.

## 3.2  Meteor Radar vs ACE v.5.3

The Institute of Geology and Geophysics of the Chinese Academy of Science has five meteor radar stations in China providing horizontal wind profiles: Mohe at $52.5°$ N, $122.3°$ E; Beijing at $40.3°$ N, $116.2°$ E; Wuhan at $30.5°$ N, $114.6°$ E; Ledong at $18.7°$ N, $109.2°$ E; and Sanya at $18.3°$ N, $109.6°$ E. The stations use VHF radar and five receivers to track the motion of meteoric trails. The measurements are reported hourly and provide an all-sky coverage (Tang et al., 2021). A detailed comparison of meteor radar data with ACE v.5.2 is provided in Johnson et al. (2024). A comparison with ACE v.5.3 is provided here, using a similar process. The spatial range for a coincidence is the same as used with MIGHTI: $2.5°$ latitude and $5°$ longitude in location and 2 hours in time.

An individual comparison between an ACE sunrise and sunset occultation with the corresponding meteor radar measurements is shown in Fig. 6. These occultations were chosen to be shown as the coincident meteor radar measurement covers a large altitude range, whereas some only provide data in a small altitude window. We again only show the meteor radar winds converted using the v.5.3 heading angles. There is excellent profile agreement for both the sunrise and sunset comparisons.

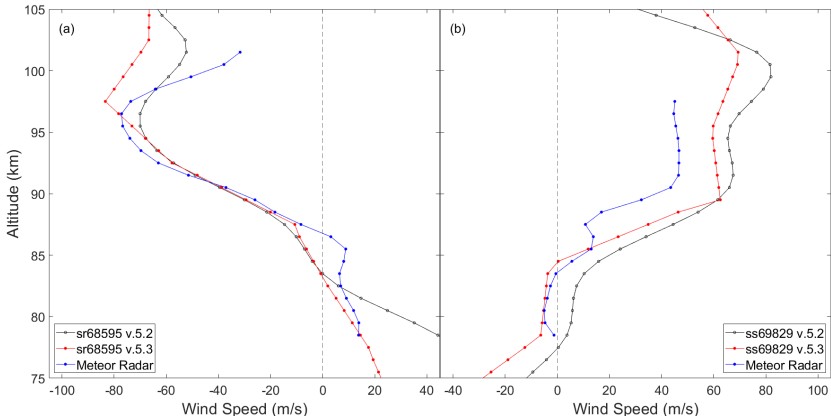

**Figure 6.** Comparison between individual ACE v.5.2 and v.5.3 **(a)** sunrise and **(b)** sunset coincidences with meteor radar measurements. Meteor radar wind speeds were converted using the ACE v.5.3 heading angle.

Unlike the comparison with MIGHTI, the number of coincidences here is small, so the entirety of the coincident data set is considered. This means, unlike in MIGHTI where we removed outliers created by measurements with small altitude windows, outliers persist towards the edges of the meteor radar altitudes. This is best seen at 100 km in Fig. 7b, where the difference in meteor radar versus ACE is nearly $40\,\mathrm{m/s}$ at the top of the window. There were 26 sunrise and 12 sunset coincidences. Of those, 18 were from the Mohe station, 12 from Beijing, and 4 from Wuhan and Ledong each. The averages of these coincidences are shown in Fig. 7. For the sunrises, we again see well matching profiles. The sunsets we see general agreement but with ACE showing more prominent features.

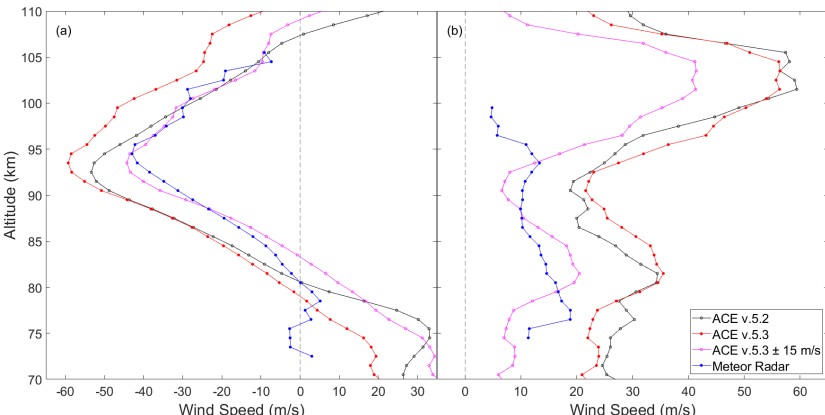

**Figure 7.** Average wind speeds of **(a)** 26 sunrise and **(b)** 12 sunset ACE v.5.2 and v.5.3 coincidences with meteor radar measurements. Meteor radar wind speeds were converted using the ACE v.5.3 heading angles.

Similar to Fig. 5 for MIGHTI, we display the average sunrise-sunset bias when comparing with meteor radar in Fig. 8. Focusing on the center of the altitude window, the sunrise (sunset) bias for ACE v.5.2 is within $\pm 2$ of $-9\,\mathrm{m/s}$ ($\pm 3$ of $+18\,\mathrm{m/s}$). For v.5.3, the bias is the same but with less variability, within $\pm 1\,\mathrm{m/s}$ ($\pm 2\,\mathrm{m/s}$).

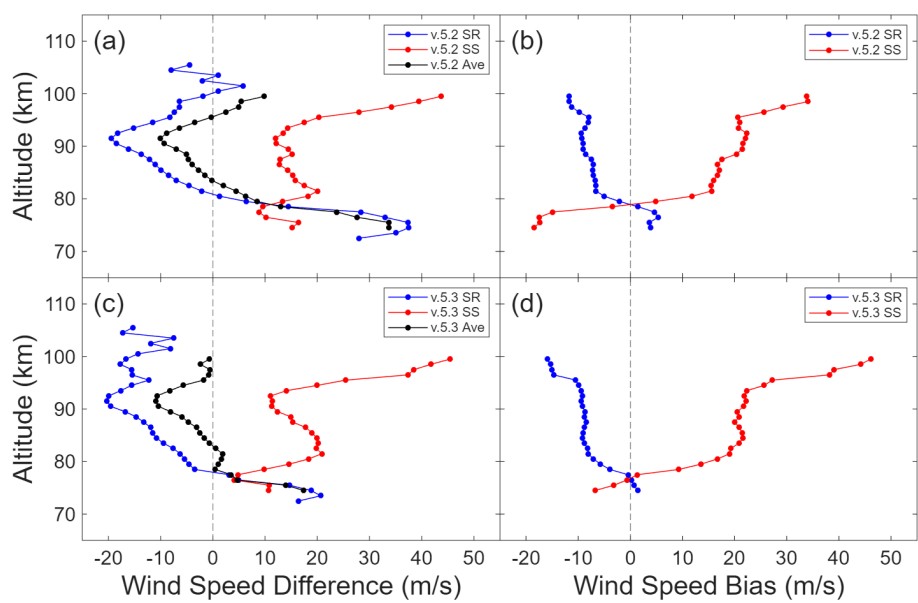

**Figure 8. (a)** Average difference between ACE v.5.2 (sunrise (blue), sunset (red), and both (black)) and meteor radar winds for each ACE wind speed measurement in 2019. **(b)** ACE sunrise-sunset bias, shown as the difference between sunrise and sunset averages and the total average in (a). Plots **(c)** and **(d)** are the same as (a) and (b), respectively, but for ACE v.5.3 processing.

## 4   Comparisons with Models

### 4.1   MERRA-2 vs ACE

MERRA-2 provides instantaneous 3-dimensional 3-hourly horizontal vector winds. The model uses cubed-sphere horizontal discretizations, giving an approximate resolution of $0.5° \times 0.625°$ and 72 hybrid-eta (altitude) levels that are best constrained up to $\sim 40\,\mathrm{km}$, but extend up to $\sim 70\,\mathrm{km}$ (Putman and Lin, 2007). Because the model is a reanalysis, we are able to compare individual ACE occultations as well as an average of measurements with the corresponding MERRA-2 data. To do this, we find the nearest MERRA-2 time to the ACE occultation. Then, we bilinearly interpolate the nearest coordinates of MERRA-2 vector winds to the given latitude and longitude of the ACE occultation. The MERRA-2 vector winds are then converted to line-of-sight winds. We again only display the MERRA-2 profiles that are calculated using the v.5.3 heading angles.

A sample of individual sunrise and sunset occultation comparisons at northern, tropical, and southern latitudes can be seen in Fig. 9. These occultations were chosen as they show particularly strong altitude profile agreement, however, other occultations show similar results. We see a decrease in bias for both sunrise and sunset occultations in v.5.3 compared with v.5.2 as well

as better matching profile. These improvements are best seen below $\sim 40$ km, where MERRA-2 is better constrained. Near 52 km in Fig. 9(a), there is a sharp eastward increase in ACE v.5.3 wind speed. This shift occurs at the boundary of segments 2 and 3 and is likely an issue with the retrieval at the top of segment 2 or bottom of segment 3 (see Fig. 1) for this particular occultation.

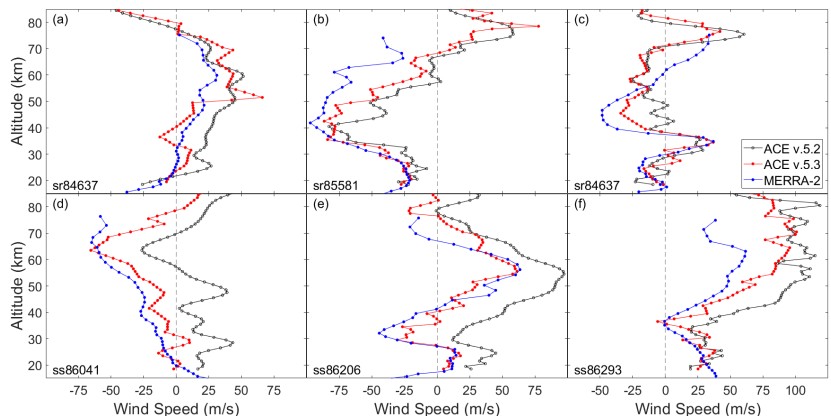

**Figure 9.** Individual ACE line-of-sight wind speeds compared with the corresponding MERRA-2 wind speeds. Plots **(a-c)** display sunrise comparisons taken during 2019 in the northern, tropical, and southern latitude regions, respectively. Plots **(d-f)** display sunset comparisons in the same regions above. **(a)** sr86572 taken at $46°$ N on September 7th, 2019. **(b)** sr84637 taken at $1°$ N on April 29th, 2019. **(c)** sr85581 taken at $45°$ S on July 2nd, 2019. **(d)** ss86041 taken at $45°$ N on August 2nd, 2019. **(e)** ss86206 taken at $1°$ S on August 13th, 2019. **(f)** ss86293 taken at $45°$ S on August 19th, 2019.

All ACE line-of-sight wind speed measurements in 2019 and their corresponding MERRA-2 wind speeds are averaged and compared in Fig. 10. Note that there are $\sim 10$ m/s jumps in MERRA-2 wind speeds, best seen between 40 and 60 km. These are not real and are due to each MERRA-2 profile having data available at varying altitudes. However, the profile produced is real. There is a large profile disagreement between 40 and 60 km for ACE v.5.2 sunrises that now shows good agreement in v.5.3. Comparing the sunsets, we see similar profile agreement but a decrease in bias, particularly at lower altitudes.

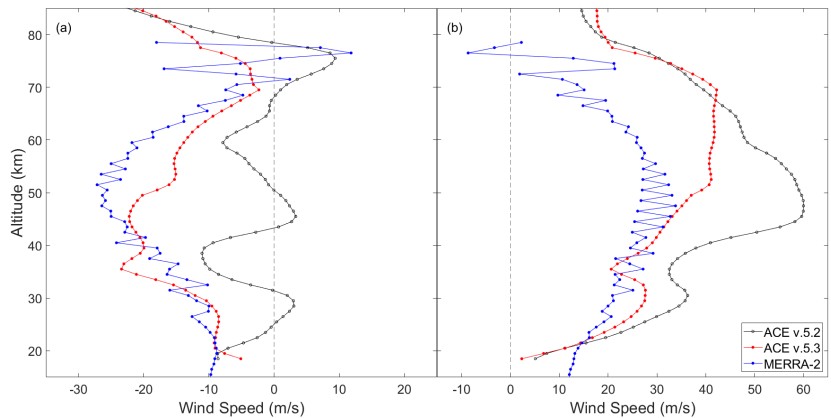

**Figure 10. (a)** Average wind speed profiles from all ACE sunrise measurements in 2019 for ACE v.5.2 (3535 occultations), ACE v.5.3 (3535 occultations), and MERRA-2. **(b)** Average wind speed profiles from all ACE sunset measurements in 2019 for ACE v.5.2 (3419 occultations), ACE v.5.3 (3422 occultations), and MERRA-2. MERRA-2 wind speeds were converted using the ACE v.5.3 heading angles. Note that the sharp jumps in MERRA-2 wind speeds are a product of our processing of the MERRA-2 data, which have varying altitudes in each vertical profile.

Similar to MIGHTI and meteor radar, we are able to derive the sunrise-sunset bias, displayed in Fig. 11. We find that MERRA-2 and ACE differ by less than 10 m/s up to 50 km with v.5.3, which is about half of the difference with v.5.2. We also find that the sunrise (sunset) bias has strongly improved. It is now within +5 m/s (−5 m/s) up through 50 km, but largely within +3 m/s (−3 m/s). Above 50 km the bias does grow to about +15 m/s (−15 m/s), as seen in v.5.2. However, this higher altitude region is not well constrained within the MERRA-2 reanalysis.

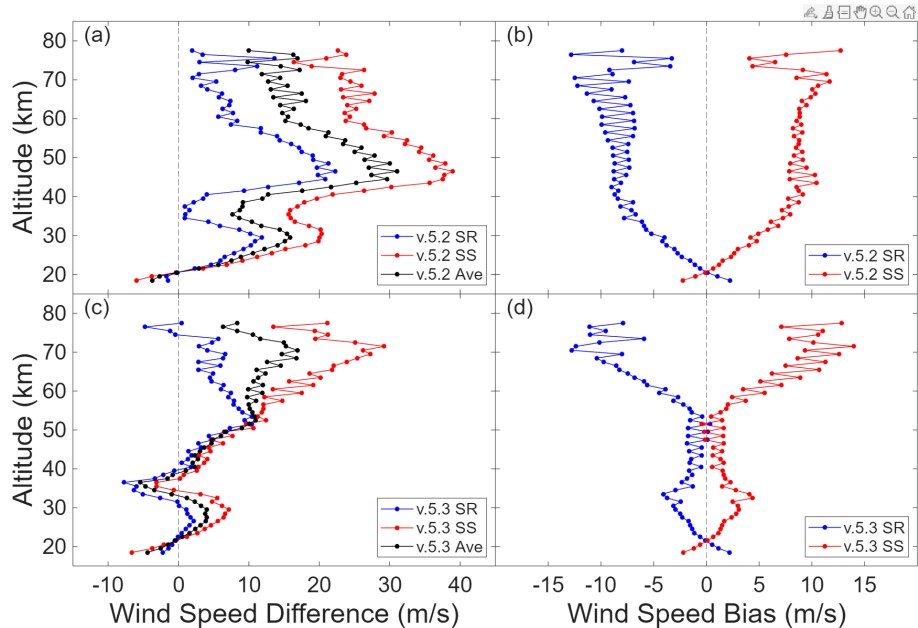

**Figure 11. (a)** Average difference between ACE v.5.2 (sunrise (blue), sunset (red), and both (black)) and MERRA-2 winds for each ACE wind speed measurement in 2019. **(b)** ACE sunrise-sunset bias, shown as the difference between sunrise and sunset averages and the total average in (a). Plots **(c)** and **(d)** are the same as (a) and (b), respectively, but for ACE v.5.3 processing.

### 4.1.1 Zonal Winds

Twice a year, around the equinoxes, the line-of-sight is such that the ACE satellite measures the east-west (or west-east) wind component, called the zonal wind. A substantial number of ACE occultations with a line-of-sight within plus or minus $10°$ of east-west (or west-east) can serve to map the zonal wind. For the year 2019, 427 sunrise occultations from March 1st ($17°$ S, heading $278°$) to April 5th ($82°$ S, heading $260°$) and 483 sunset occultations from September 6th ($81°$ S, heading $100°$) to October 19th ($41°$ N, heading $80°$) fall within the $\pm10°$ zonal wind category in the southern hemisphere. These occultations are more densely distributed near the pole. For $2°$ latitude bins, all wind profiles were averaged and the resulting maps of the $\pm10°$ zonal wind intensity as a function of altitude and latitude are shown in Fig. 12(a) and (c). The March-April wind signs were inverted so that positive zonal winds (in red) indicate winds moving from west to east. Note that the heading angle is exactly $270°$ (respectively $90°$) for latitude around $73°$ S ($71°$ S), which correspond to March 20th (September 23rd).

ACE measures the polar vortex near 30 km altitude in September-October. As expected, the vortex is weak in March-April. In September-October, the mesosphere is characterized by strong positive zonal winds in the whole hemisphere. It is well known that winds of the thermosphere are variable with local time. Because of ACE observation geometry, the sampled local times for sunrises (March-April) are typically between 6:00 and 9:00 and the sampled local times for sunsets (September-October) are typically between 15:00 and 18:00. For sunrises in March-April, the thermosphere zonal winds are positive around 100 km altitude and negative above. At sunsets in September-October, strong negative zonal winds are observed at $60°$ S and

altitude 130 km. Notably, the winds in the tropics are in opposite directions at 50 km (eastward) and 80 km (westward) in September-October. This is representative of the vertical structure of the semiannual oscillation (Ern et al., 2021).

MERRA-2 reanalysis wind vectors were sampled for each of the occultations and projected into the corresponding ACE line-of-sight direction. The latitude-bin-averaged MERRA-2 winds are shown in Fig. 12(b) and (d). For the September-October period, the MERRA-2 winds below 40 km are in good agreement with ACE, with errors lower than 15 m/s. Above 40 km, the MERRA-2 winds are weaker by about 20 m/s compared to ACE. For the March-October period, the MERRA-2 winds mostly agree with the ACE winds with errors less than 10 m/s. The strength of the negative zonal winds around $20°$ S and altitude 40 km is overestimated by MERRA-2 by around 15 m/s.

Overall, the ACE data show sharper wind features than MERRA-2 in the stratosphere and an intriguing wave-like pattern spanning $80°$ S to $20°$ S in both time periods. This pattern could be an artifact due to the smooth varying line-of-sight angle with latitude combined with winds in the non-tangent layers. At altitudes above 80 km, ACE provides a unique measurement of the complex dynamics of the thermosphere.

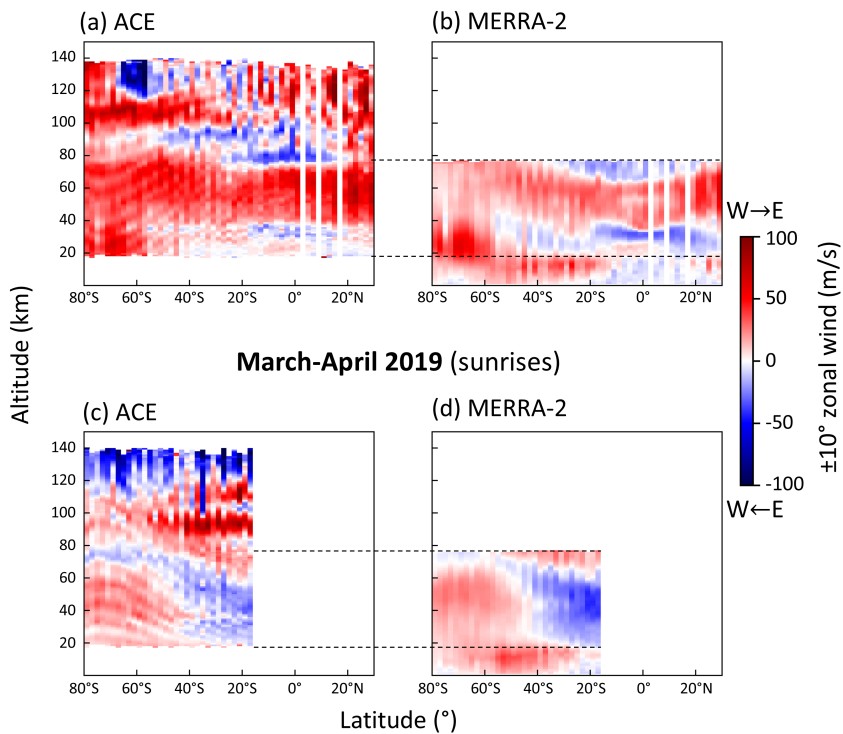

**Figure 12.** ACE measurements of the $\pm10°$ zonal winds as a function of latitude and altitude **(a, c)** and corresponding MERRA-2 winds **(b, d)** for year 2019. The September-October observations (a, b) correspond to sunset solar occultations with local time at the tangent point between 6:00 am and 9:00 am and the March-April observations (c, d) correspond to sunrise solar occultations with local time at the tangent point between 15:00 pm and 18:00 pm.

## 4.2 HWM14 vs ACE

HWM14 outputs for all of 2019 on a $5° \times 5°$ grid with 15 minute time increments using observational geomagnetic ap indexes were retrieved from the Community Coordinated Modeling Center (CCMC) based out of NASA Goddard Space Flight Center (GSFC). The data ranges from the surface up to 1000 km in 10 km intervals.

Since HWM14 is a climatology, we only compare the average of many ACE and corresponding HWM14 wind speeds. We again compare all ACE wind speed measurements available from 2019. We first separate the ACE occultations into northern (latitude $> 45°$ N), tropical (latitude between $30°$ S and $30°$ N), and southern (latitude $< 45°$ S) regions based on latitude. It is not reasonable for the climatology to predict wind patterns within the winter polar vortex, so measurements during those periods are removed. Similar to MERRA-2, we then find the nearest HWM14 time to a given ACE occultation and bilinearly interpolate the vector winds of the nearby HWM14 coordinates to the given ACE coordinates. The HWM14 vector winds are then converted to line-of-sight winds.

This average of ACE v.5.2 and v.5.3 wind speeds from 2019 and their corresponding HWM14 wind speeds are shown in
Fig. 13. For v.5.2 (v.5.3), 1037 (1032), 410 (418), and 945 (945) sunrise occultations were used for northern, tropical, and
southern latitudes. For sunsets, 1300 (1294), 396 (402), and 924 (923) were used, respectively. The displayed HWM14 wind
speeds are calculated using the v.5.3 heading angles. We see general profile agreement for both processing versions, with v.5.3
being slightly improved. Both versions of ACE processing show more prominent features than those found in HWM14, best
seen at the $\sim 90$ and $\sim 60$ km features in Fig. 13(a) and (e), respectively. The sunrise-sunset bias relative to HWM14 is shown
in Fig. 14, similar to Fig. 11 relative to MERRA-2. We see a similar shape to that of MERRA-2, but the bias in v.5.2 appears
less significant than in v.5.3. At low altitudes, the bias begins at less than 5 m/s, but increases to 17 m/s at 60 km for v.5.2 and
20 m/s at 70 km for v.5.3. In then decreases through 90 km before rising to it's maximum of 22 m/s and 37 m/s at 100 km
for v.5.2 and v.5.3, respectively.

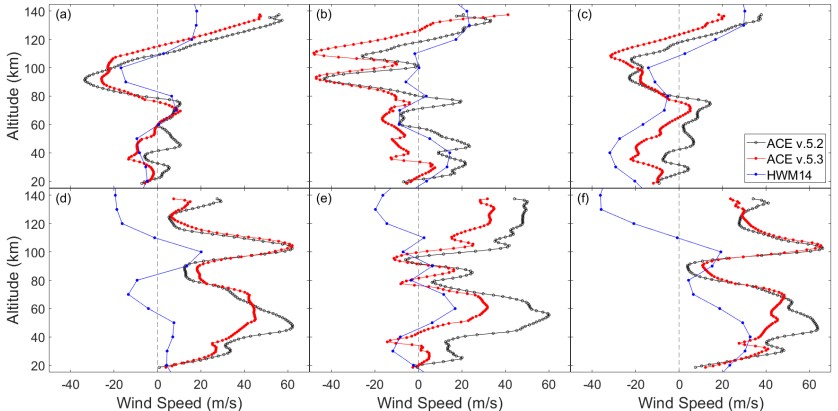

**Figure 13.** Average line-of-sight wind speeds from ACE measurements in 2019 for ACE v.5.2, ACE v.5.3, and HWM14. HWM14 wind speeds were converted using the ACE v.5.3 heading angles. **(a-c)** are from sunrise measurements in the northern (latitude $> 45°$ N), tropical (latitude between $30°$ S and $30°$ N), and southern (latitude $< 45°$ S) regions, respectively. **(d-f)** match the same regions above, but for sunset measurements.

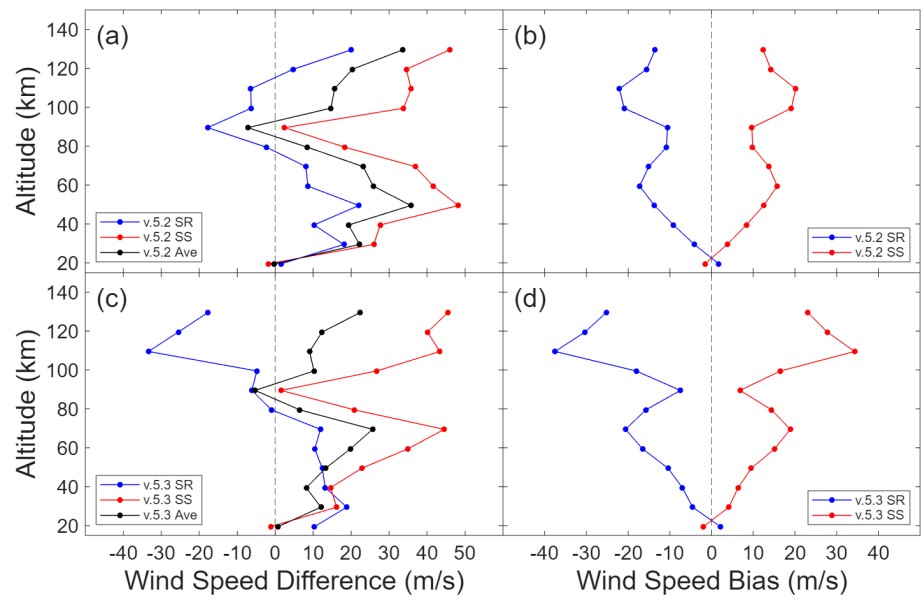

**Figure 14. (a)** Average difference between ACE v.5.2 (sunrise (blue), sunset (red), and both (black)) and HWM14 winds for each ACE wind speed measurement in 2019. **(b)** ACE sunrise-sunset bias, shown as the difference between sunrise and sunset averages and the total average in (a). Plots **(c)** and **(d)** are the same as (a) and (b), respectively, but for ACE v.5.3 processing.

## 4.3 WACCM-X vs ACE

Specified Dynamics (SD) WACCM-X Version 2.2 provides global vector winds in 3 hour intervals on a $1.9° \times 2.5°$ grid. This SD version of WACCM-X uses observational data to produce wind speeds closer to the actual atmospheric state. WACCM-X vector winds were compared with ACE-FTS line-of-sight winds for all of 2019 in a similar fashion as HWM14. The resulting line-of-sight winds are shown in Fig. 15. There is notably less agreement between both ACE versions and the WACCM-X profiles compared with HWM14. Both ACE sunrise and sunset averages for v.5.2 and v.5.3 in the northern and southern latitude

region show additional features not found in the WACCM-X model. The model and ACE data show the best profile agreement in the tropical region. Comparing the sunrise-sunset biases, shown in Fig. 16(b) and (d) for v.5.2 and v.5.3, respectively, we see a similar shape as before when compared with HWM14. The sunrise (sunset) bias varies up through 80 km, but is less than $-8 \, \text{m/s} \, (+7 \, \text{m/s})$. It then increases drastically to $-34 \, \text{m/s} \, (+32 \, \text{m/s})$ at 103 km before dropped to near zero at 120 km.

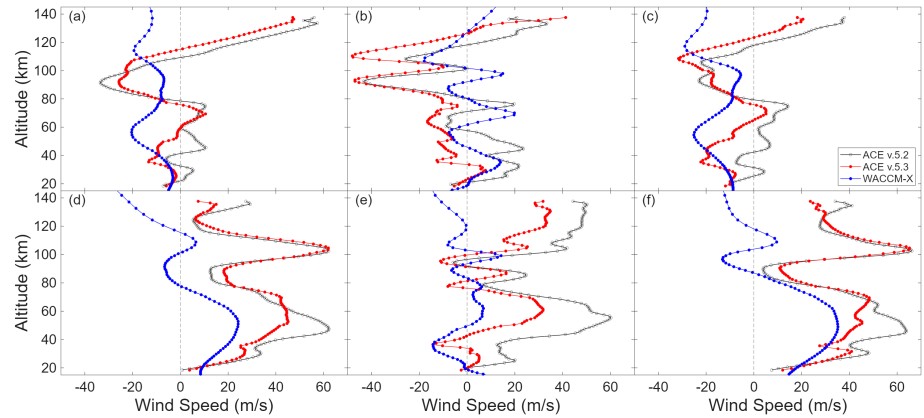

**Figure 15.** Average line-of-sight wind speeds from ACE measurements in 2019 for ACE v.5.2 (black), ACE v.5.3 (red), and WACCM-X (blue). WACCM-X wind speeds were converted using the ACE v.5.3 heading angles. **(a-c)** are from sunrise measurements in the northern (latitude $> 45°$ N), tropical (latitude between $30°$ S and $30°$ N), and southern (latitude $< 45°$ S) regions, respectively. **(d-f)** match the same regions above, but for sunset measurements.

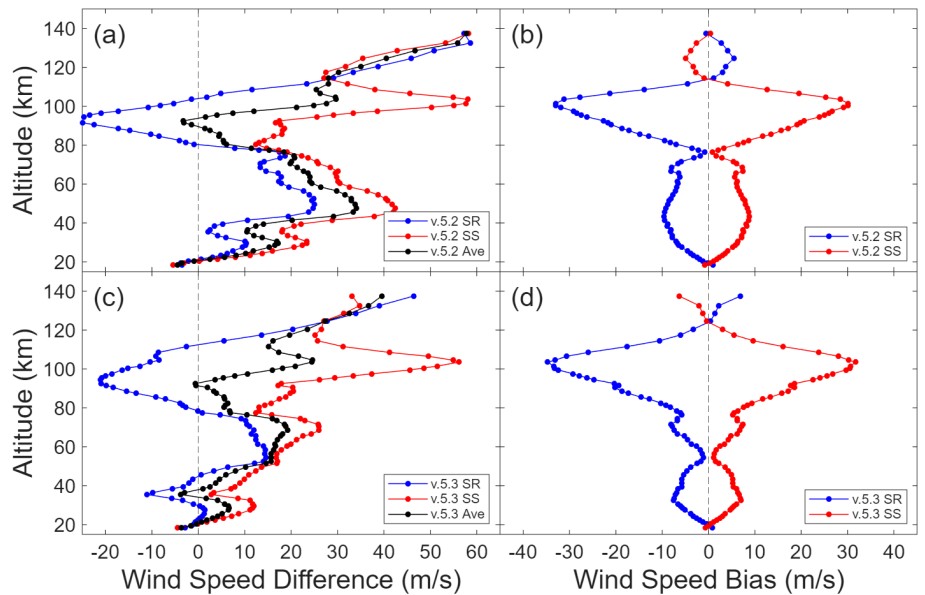

**Figure 16. (a)** Average difference between ACE v.5.2 (sunrise (blue), sunset (red), and both (black)) and WACCM-X winds for each ACE wind speed measurement in 2019. **(b)** ACE sunrise-sunset bias, shown as the difference between sunrise and sunset averages and the total average in (a). Plots **(c)** and **(d)** are the same as (a) and (b), respectively, but for ACE v.5.3 processing.

## 5   Conclusions

ACE v.5.3 wind speeds have been validated by instrument observations from MIGHTI and meteor radar. Wind speed profiles from ACE v.5.3 for sunrise and sunset occultations show improved profile agreement with MIGHTI and meteor radar measurements. The approximate $-15$ m/s ($+15$ m/s) sunrise (sunset) bias previously found is still seen above 80 km.

ACE v.5.2 and v.5.3 line-of-sight wind speeds have been compared with vector winds from the MERRA-2, HWM14, and WACCM-X models. The new wind speeds have better profile agreement than v.5.2 does with each model. The v.5.3 processing
has particularly strong profile agreement with MERRA-2 below 40 km, where the model is well constrained. The MERRA-2 comparison shows a decreased sunrise (sunset) bias of less than $-5$ m/s ($+5$ m/s) below 50 km. At higher altitudes, the bias is nearer to $-15$ to $-20$ m/s ($+15$ to $+20$ m/s), with a sharp increase close to 100 km.

The ACE-FTS instrument on the Canadian SCISAT satellite now provides line-of-sight wind speeds between 18 and 140 km dating back to 2004 with v.5.2 and the improved v.5.3 processing. There is a lack of wind speed measurements within this
altitude region from other sources, making ACE wind speeds a valuable addition for model evaluation.

*Author contributions.*   **MW:** Formal analysis, Writing - original draft; **PB:** Supervision, Writing - review and editing; **CB:** Investigation, Writing - original draft; **LL:** Formal analysis, Writing - original draft; **RJ:** Investigation.

*Data availability.*   ACE v.5.2 and v.5.3 wind data can be found on the SCISAT webpage (https://databace.scisat.ca) within the Level 2 Data.

*Competing interests.*   The authors declare that they have no conflict of interest.

*Acknowledgements.*   PB acknowledges RB for productive discussion.

*Financial support.*   The ACE mission is funded by the Canadian Space Agency (9F045-230412/001). Some support was provided by NASA ACMAP, Atmospheric Composition Modeling and Analysis Program (80NSSC23K0999).

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

# Appendix A

| Altitude Segment | Center (cm$^{-1}$) | Width (cm$^{-1}$) | Altitude Range (km) | Transmittance Range | Isotopologue |
|---|---|---|---|---|---|
| 1 | 2602.30 | 0.4 | 18-35 | 0.4-0.97 | $^{18}OCO$ |
| 1 | 2604.50 | 0.8 | 18-35 | 0.4-0.97 | $^{18}OCO$ |
| 1 | 2606.00 | 0.8 | 18-35 | 0.4-0.97 | $^{18}OCO$ |
| 1 | 2607.00 | 0.8 | 18-35 | 0.4-0.97 | $^{18}OCO$ |
| 1 | 2608.00 | 1.0 | 18-35 | 0.4-0.97 | $^{18}OCO$ |
| 1 | 2609.80 | 0.4 | 18-35 | 0.4-0.97 | $^{18}OCO$ |
| 1 | 2610.50 | 1.0 | 18-35 | 0.4-0.97 | $^{18}OCO$ |
| 1 | 2610.90 | 1.0 | 18-35 | 0.4-0.97 | $^{18}OCO$ |
| 1 | 2611.60 | 1.0 | 18-35 | 0.4-0.97 | $^{18}OCO$ |
| 1 | 2612.40 | 1.0 | 18-35 | 0.4-0.97 | $^{18}OCO$ |
| 1 | 2616.10 | 1.0 | 18-35 | 0.4-0.97 | $^{18}OCO$ |
| 1 | 2616.84 | 1.0 | 18-35 | 0.4-0.97 | $^{18}OCO$ |
| 1 | 2617.50 | 1.0 | 18-35 | 0.4-0.97 | $^{18}OCO$ |
| 1 | 2619.01 | 0.9 | 18-35 | 0.4-0.97 | $^{18}OCO$ |
| 1 | 2620.50 | 1.0 | 18-35 | 0.4-0.97 | $^{18}OCO$ |
| 1 | 2621.25 | 1.1 | 18-35 | 0.4-0.97 | $^{18}OCO$ |
| 1 | 2623.40 | 1.0 | 18-35 | 0.4-0.97 | $^{18}OCO$ |
| 1 | 2624.80 | 1.0 | 18-35 | 0.4-0.97 | $^{18}OCO$ |
| 1 | 2627.00 | 1.0 | 18-35 | 0.4-0.97 | $^{18}OCO$ |
| 1 | 2629.80 | 1.0 | 18-35 | 0.4-0.97 | $^{18}OCO$ |
| 1 | 2632.05 | 1.0 | 18-35 | 0.4-0.97 | $^{18}OCO$ |
| 1 | 2636.30 | 1.0 | 18-35 | 0.4-0.97 | $^{18}OCO$ |
| 2 | 1889.02 | 0.4 | 30-45 | 0.65-0.96 | $CO_2$ |
| 2 | 1891.97 | 0.3 | 30-50 | 0.65-0.96 | $CO_2$ |
| 2 | 1896.17 | 0.3 | 30-55 | 0.65-0.96 | $CO_2$ |
| 2 | 1897.69 | 0.3 | 30-55 | 0.65-0.96 | $CO_2$ |
| 2 | 1898.13 | 0.3 | 30-45 | 0.65-0.96 | $CO_2$ |
| 2 | 1899.17 | 0.3 | 35-60 | 0.65-0.96 | $CO_2$ |
| 2 | 1899.29 | 0.2 | 30-45 | 0.65-0.96 | $CO_2$ |
| 2 | 1902.15 | 0.3 | 35-60 | 0.65-0.96 | $CO_2$ |
| 2 | 1905.05 | 0.3 | 35-60 | 0.65-0.96 | $CO_2$ |
| 2 | 1905.37 | 0.3 | 30-45 | 0.65-0.96 | $CO_2$ |
| 2 | 1906.43 | 0.3 | 35-60 | 0.65-0.96 | $CO_2$ |

| Altitude Segment | Center (cm$^{-1}$) | Width (cm$^{-1}$) | Altitude Range (km) | Transmittance Range | Isotopologue |
|---|---|---|---|---|---|
| 2 | 1908.37 | 0.3 | 30-45 | 0.65-0.96 | $CO_2$ |
| 2 | 1909.45 | 0.3 | 35-60 | 0.65-0.96 | $CO_2$ |
| 2 | 1911.01 | 0.3 | 35-60 | 0.65-0.96 | $CO_2$ |
| 2 | 1912.53 | 0.3 | 45-60 | 0.65-0.96 | $CO_2$ |
| 2 | 1914.11 | 0.3 | 35-60 | 0.65-0.96 | $CO_2$ |
| 2 | 1915.47 | 0.3 | 35-60 | 0.65-0.96 | $CO_2$ |
| 2 | 1917.05 | 0.3 | 35-60 | 0.65-0.96 | $CO_2$ |
| 2 | 1920.11 | 0.3 | 35-60 | 0.65-0.96 | $CO_2$ |
| 2 | 1924.77 | 0.3 | 35-60 | 0.65-0.96 | $CO_2$ |
| 2 | 1927.81 | 0.3 | 30-60 | 0.65-0.96 | $CO_2$ |
| 2 | 1929.35 | 0.3 | 30-60 | 0.65-0.96 | $CO_2$ |
| 2 | 1930.85 | 0.3 | 30-50 | 0.65-0.96 | $CO_2$ |
| 2 | 1933.91 | 0.3 | 30-60 | 0.65-0.96 | $CO_2$ |
| 2 | 1935.16 | 0.4 | 30-60 | 0.65-0.96 | $CO_2$ |
| 2 | 1936.47 | 0.3 | 30-55 | 0.65-0.96 | $CO_2$ |
| 2 | 1941.05 | 0.3 | 30-50 | 0.65-0.96 | $CO_2$ |
| 2 | 1950.67 | 0.3 | 30-55 | 0.65-0.96 | $CO_2$ |
| 2 | 1966.85 | 0.3 | 30-55 | 0.65-0.96 | $CO_2$ |
| 2 | 1970.31 | 0.3 | 30-55 | 0.65-0.96 | $CO_2$ |
| 2 | 1971.77 | 0.3 | 30-50 | 0.65-0.96 | $CO_2$ |
| 2 | 1975.05 | 0.3 | 35-45 | 0.65-0.96 | $CO_2$ |
| 3 | 2039.93 | 0.3 | 50-77 | 0.65-0.97 | $CO_2$ |
| 3 | 2042.95 | 0.3 | 50-77 | 0.65-0.97 | $CO_2$ |
| 3 | 2044.45 | 0.3 | 50-77 | 0.65-0.97 | $CO_2$ |
| 3 | 2045.95 | 0.3 | 50-77 | 0.65-0.97 | $CO_2$ |
| 3 | 2047.51 | 0.3 | 50-77 | 0.65-0.97 | $CO_2$ |
| 3 | 2049.03 | 0.3 | 50-77 | 0.65-0.97 | $CO_2$ |
| 3 | 2050.55 | 0.3 | 50-77 | 0.65-0.97 | $CO_2$ |
| 3 | 2052.09 | 0.3 | 50-77 | 0.65-0.97 | $CO_2$ |
| 3 | 2053.57 | 0.3 | 50-77 | 0.65-0.97 | $CO_2$ |
| 3 | 2055.15 | 0.3 | 50-77 | 0.65-0.97 | $CO_2$ |
| 3 | 2056.69 | 0.3 | 50-77 | 0.65-0.97 | $CO_2$ |
| 3 | 2058.29 | 0.3 | 50-77 | 0.65-0.97 | $CO_2$ |

| Altitude Segment | Center (cm$^{-1}$) | Width (cm$^{-1}$) | Altitude Range (km) | Transmittance Range | Isotopologue |
|---|---|---|---|---|---|
| 3 | 2059.73 | 0.3 | 50-77 | 0.65-0.97 | $CO_2$ |
| 3 | 2061.25 | 0.3 | 50-77 | 0.65-0.97 | $CO_2$ |
| 3 | 2062.87 | 0.3 | 50-77 | 0.65-0.97 | $CO_2$ |
| 3 | 2066.01 | 0.3 | 60-77 | 0.65-0.97 | $CO_2$ |
| 3 | 2067.51 | 0.3 | 60-77 | 0.65-0.97 | $CO_2$ |
| 3 | 2070.61 | 0.3 | 60-77 | 0.65-0.97 | $CO_2$ |
| 3 | 2072.17 | 0.3 | 60-77 | 0.65-0.97 | $CO_2$ |
| 4 | 2232.95 | 0.3 | 70-80 | 0.67-0.975 | $^{13}CO_2$ |
| 4 | 2237.27 | 0.3 | 70-90 | 0.67-0.975 | $^{13}CO_2$ |
| 4 | 2239.39 | 0.3 | 70-90 | 0.67-0.975 | $^{13}CO_2$ |
| 4 | 2243.59 | 0.3 | 70-95 | 0.67-0.975 | $^{13}CO_2$ |
| 4 | 2245.65 | 0.3 | 70-95 | 0.67-0.975 | $^{13}CO_2$ |
| 4 | 2248.71 | 0.3 | 70-85 | 0.67-0.975 | $^{13}CO_2$ |
| 4 | 2249.66 | 0.3 | 70-95 | 0.67-0.975 | $^{13}CO_2$ |
| 4 | 2250.49 | 0.3 | 70-85 | 0.67-0.975 | $^{13}CO_2$ |
| 4 | 2251.35 | 0.3 | 70-85 | 0.67-0.975 | $^{13}CO_2$ |
| 4 | 2251.65 | 0.3 | 70-95 | 0.67-0.975 | $^{13}CO_2$ |
| 4 | 2252.31 | 0.3 | 70-85 | 0.67-0.975 | $^{13}CO_2$ |
| 4 | 2253.19 | 0.3 | 70-85 | 0.67-0.975 | $^{13}CO_2$ |
| 4 | 2253.75 | 0.3 | 70-100 | 0.67-0.975 | $^{13}CO_2$ |
| 4 | 2254.09 | 0.3 | 70-85 | 0.67-0.975 | $^{13}CO_2$ |
| 4 | 2254.91 | 0.3 | 70-85 | 0.67-0.975 | $^{13}CO_2$ |
| 4 | 2255.59 | 0.3 | 70-105 | 0.67-0.975 | $^{13}CO_2$ |
| 4 | 2257.60 | 0.4 | 85-105 | 0.67-0.975 | $^{13}CO_2$ |
| 4 | 2260.18 | 0.3 | 70-85 | 0.67-0.975 | $^{13}CO_2$ |
| 4 | 2261.20 | 0.4 | 85-110 | 0.67-0.975 | $^{13}CO_2$ |
| 4 | 2261.93 | 0.3 | 70-85 | 0.67-0.975 | $^{13}CO_2$ |
| 4 | 2262.75 | 0.3 | 70-85 | 0.67-0.975 | $^{13}CO_2$ |
| 4 | 2263.21 | 0.3 | 70-110 | 0.67-0.975 | $^{13}CO_2$ |
| 4 | 2264.95 | 0.3 | 70-110 | 0.67-0.975 | $^{13}CO_2$ |
| 4 | 2266.07 | 0.3 | 70-85 | 0.67-0.975 | $^{13}CO_2$ |
| 4 | 2268.53 | 0.3 | 85-110 | 0.67-0.975 | $^{13}CO_2$ |
| 4 | 2270.29 | 0.3 | 90-110 | 0.67-0.975 | $^{13}CO_2$ |

430

| Altitude Segment | Center (cm$^{-1}$) | Width (cm$^{-1}$) | Altitude Range (km) | Transmittance Range | Isotopologue |
|---|---|---|---|---|---|
| 4 | 2272.03 | 0.3 | 70-110 | 0.67-0.975 | $^{13}CO_2$ |
| 4 | 2273.73 | 0.3 | 90-110 | 0.67-0.975 | $^{13}CO_2$ |
| 4 | 2275.41 | 0.3 | 70-110 | 0.67-0.975 | $^{13}CO_2$ |
| 4 | 2277.07 | 0.3 | 90-110 | 0.67-0.975 | $^{13}CO_2$ |
| 4 | 2278.60 | 0.4 | 80-110 | 0.67-0.975 | $^{13}CO_2$ |
| 4 | 2280.33 | 0.3 | 80-110 | 0.67-0.975 | $^{13}CO_2$ |
| 4 | 2281.85 | 0.3 | 85-110 | 0.67-0.975 | $^{13}CO_2$ |
| 4 | 2284.33 | 0.3 | 80-110 | 0.67-0.975 | $^{13}CO_2$ |
| 4 | 2285.86 | 0.3 | 80-110 | 0.67-0.975 | $^{13}CO_2$ |
| 4 | 2287.33 | 0.3 | 90-110 | 0.67-0.975 | $^{13}CO_2$ |
| 4 | 2288.79 | 0.3 | 85-110 | 0.67-0.975 | $^{13}CO_2$ |
| 4 | 2290.25 | 0.3 | 90-110 | 0.67-0.975 | $^{13}CO_2$ |
| 4 | 2291.79 | 0.3 | 80-110 | 0.67-0.975 | $^{13}CO_2$ |
| 4 | 2293.09 | 0.3 | 90-110 | 0.67-0.975 | $^{13}CO_2$ |
| 4 | 2294.53 | 0.3 | 90-110 | 0.67-0.975 | $^{13}CO_2$ |
| 4 | 2299.87 | 0.3 | 90-110 | 0.67-0.975 | $^{13}CO_2$ |
| 4 | 2302.29 | 0.3 | 90-110 | 0.67-0.975 | $^{13}CO_2$ |
| 4 | 2303.47 | 0.3 | 90-110 | 0.67-0.975 | $^{13}CO_2$ |
| 5 | 2300.50 | 0.4 | 95-120 | 0.67-0.985 | $CO_2$ |
| 5 | 2302.65 | 0.3 | 95-120 | 0.67-0.985 | $CO_2$ |
| 5 | 2306.95 | 0.3 | 95-140 | 0.67-0.985 | $CO_2$ |
| 5 | 2309.03 | 0.3 | 100-140 | 0.67-0.985 | $CO_2$ |
| 5 | 2309.63 | 0.3 | 95-105 | 0.67-0.985 | $CO_2$ |
| 5 | 2311.11 | 0.3 | 105-140 | 0.67-0.985 | $CO_2$ |
| 5 | 2311.59 | 0.3 | 95-105 | 0.67-0.985 | $CO_2$ |
| 5 | 2313.15 | 0.3 | 95-140 | 0.67-0.985 | $CO_2$ |
| 5 | 2314.29 | 0.3 | 95-110 | 0.67-0.985 | $CO_2$ |
| 5 | 2315.21 | 0.3 | 95-140 | 0.67-0.985 | $CO_2$ |
| 5 | 2316.07 | 0.3 | 95-115 | 0.67-0.985 | $CO_2$ |
| 5 | 2317.19 | 0.3 | 110-140 | 0.67-0.985 | $CO_2$ |
| 5 | 2319.10 | 0.4 | 95-140 | 0.67-0.985 | $CO_2$ |
| 5 | 2319.89 | 0.3 | 95-115 | 0.67-0.985 | $CO_2$ |
| 5 | 2320.75 | 0.3 | 95-115 | 0.67-0.985 | $CO_2$ |

| Altitude Segment | Center $(cm^{-1})$ | Width $(cm^{-1})$ | Altitude Range (km) | Transmittance Range | Isotopologue |
|---|---|---|---|---|---|
| 5 | 2321.11 | 0.3 | 105-140 | 0.67-0.985 | $CO_2$ |
| 5 | 2321.61 | 0.3 | 95-115 | 0.67-0.985 | $CO_2$ |
| 5 | 2322.51 | 0.3 | 95-115 | 0.67-0.985 | $CO_2$ |
| 5 | 2323.11 | 0.3 | 95-140 | 0.67-0.985 | $CO_2$ |
| 5 | 2323.37 | 0.3 | 95-115 | 0.67-0.985 | $CO_2$ |
| 5 | 2324.24 | 0.3 | 95-115 | 0.67-0.985 | $CO_2$ |
| 5 | 2325.00 | 0.4 | 95-140 | 0.67-0.985 | $CO_2$ |
| 5 | 2328.63 | 0.3 | 95-140 | 0.67-0.985 | $CO_2$ |
| 5 | 2329.39 | 0.3 | 95-115 | 0.67-0.985 | $CO_2$ |
| 5 | 2330.17 | 0.3 | 95-115 | 0.67-0.985 | $CO_2$ |
| 5 | 2330.55 | 0.3 | 105-140 | 0.67-0.985 | $CO_2$ |
| 5 | 2331.03 | 0.3 | 95-115 | 0.67-0.985 | $CO_2$ |
| 5 | 2331.85 | 0.3 | 95-115 | 0.67-0.985 | $CO_2$ |
| 5 | 2332.35 | 0.3 | 95-140 | 0.67-0.985 | $CO_2$ |
| 5 | 2332.65 | 0.3 | 95-115 | 0.67-0.985 | $CO_2$ |
| 5 | 2333.37 | 0.3 | 95-115 | 0.67-0.985 | $CO_2$ |
| 5 | 2334.15 | 0.3 | 105-140 | 0.67-0.985 | $CO_2$ |
| 5 | 2337.59 | 0.3 | 100-140 | 0.67-0.985 | $CO_2$ |
| 5 | 2338.93 | 0.3 | 95-110 | 0.67-0.985 | $CO_2$ |
| 5 | 2339.37 | 0.3 | 95-140 | 0.67-0.985 | $CO_2$ |
| 5 | 2339.65 | 0.3 | 95-115 | 0.67-0.985 | $CO_2$ |
| 5 | 2341.07 | 0.3 | 110-140 | 0.67-0.985 | $CO_2$ |
| 5 | 2343.39 | 0.3 | 95-115 | 0.67-0.985 | $CO_2$ |
| 5 | 2344.09 | 0.3 | 95-115 | 0.67-0.985 | $CO_2$ |
| 5 | 2344.37 | 0.3 | 110-140 | 0.67-0.985 | $CO_2$ |
| 5 | 2344.77 | 0.3 | 95-115 | 0.67-0.985 | $CO_2$ |
| 5 | 2346.05 | 0.3 | 95-140 | 0.67-0.985 | $CO_2$ |
| 5 | 2347.59 | 0.3 | 110-140 | 0.67-0.985 | $CO_2$ |
| 5 | 2348.31 | 0.3 | 95-115 | 0.67-0.985 | $CO_2$ |
| 5 | 2348.97 | 0.3 | 95-115 | 0.67-0.985 | $CO_2$ |
| 5 | 2349.55 | 0.3 | 95-115 | 0.67-0.985 | $CO_2$ |
| 5 | 2350.01 | 0.3 | 100-140 | 0.67-0.985 | $CO_2$ |
| 5 | 2351.45 | 0.3 | 110-140 | 0.67-0.985 | $CO_2$ |
| | | | | | *Continued on next page* |

| Altitude Segment | Center (cm$^{-1}$) | Width (cm$^{-1}$) | Altitude Range (km) | Transmittance Range | Isotopologue |
|---|---|---|---|---|---|
| 5 | 2352.31 | 0.3 | 95-115 | 0.67-0.985 | $CO_2$ |
| 5 | 2354.43 | 0.3 | 105-140 | 0.67-0.985 | $CO_2$ |
| 5 | 2355.43 | 0.3 | 95-115 | 0.67-0.985 | $CO_2$ |
| 5 | 2355.89 | 0.3 | 105-140 | 0.67-0.985 | $CO_2$ |
| 5 | 2357.31 | 0.3 | 105-140 | 0.67-0.985 | $CO_2$ |
| 5 | 2358.73 | 0.3 | 105-140 | 0.67-0.985 | $CO_2$ |
| 5 | 2361.47 | 0.3 | 100-140 | 0.67-0.985 | $CO_2$ |
| 5 | 2362.79 | 0.3 | 105-140 | 0.67-0.985 | $CO_2$ |
| 5 | 2364.11 | 0.3 | 95-140 | 0.67-0.985 | $CO_2$ |
| 5 | 2365.39 | 0.3 | 100-140 | 0.67-0.985 | $CO_2$ |
| 5 | 2366.65 | 0.3 | 95-140 | 0.67-0.985 | $CO_2$ |
| 5 | 2367.87 | 0.3 | 95-140 | 0.67-0.985 | $CO_2$ |
| 5 | 2369.09 | 0.3 | 95-140 | 0.67-0.985 | $CO_2$ |
| 5 | 2370.27 | 0.3 | 95-140 | 0.67-0.985 | $CO_2$ |
| 5 | 2371.43 | 0.3 | 95-140 | 0.67-0.985 | $CO_2$ |
| 5 | 2372.55 | 0.3 | 95-140 | 0.67-0.985 | $CO_2$ |
| 5 | 2373.67 | 0.3 | 95-140 | 0.67-0.985 | $CO_2$ |
| 5 | 2374.75 | 0.3 | 95-140 | 0.67-0.985 | $CO_2$ |
| 5 | 2375.79 | 0.3 | 95-140 | 0.67-0.985 | $CO_2$ |
| 5 | 2376.83 | 0.3 | 95-140 | 0.67-0.985 | $CO_2$ |
| 5 | 2377.85 | 0.3 | 95-140 | 0.67-0.985 | $CO_2$ |
| 5 | 2378.83 | 0.3 | 95-140 | 0.67-0.985 | $CO_2$ |
| 5 | 2379.79 | 0.3 | 95-140 | 0.67-0.985 | $CO_2$ |
| 5 | 2380.71 | 0.3 | 95-140 | 0.67-0.985 | $CO_2$ |
| 5 | 2381.61 | 0.3 | 95-140 | 0.67-0.985 | $CO_2$ |
| 5 | 2382.49 | 0.3 | 95-140 | 0.67-0.985 | $CO_2$ |
| 5 | 2383.35 | 0.3 | 95-140 | 0.67-0.985 | $CO_2$ |

**Table A1.** Microwindow sets employed in the 5 altitude segments for the wind retrieval from ACE-FTS measurements, along with the allowed range for minimum transmittance in the window used to filter saturated or excessively weak lines from the analysis.