# Peer review of "Atmospheric Chemistry Experiment (ACE) v.5.3 Winds: Validation and Model Comparisons"

_EGUsphere, 2025_

## Author Comment (AC1)

**RC1 Comments**

**(1) 1.44-51: Suggestion for future work:**

You should compare ACE LOS winds with the Japanese JAWARA reanalysis that was developed in the frame of the ICSOM project. Unlike MERRA2, JAWARA assimilates data even in the upper mesosphere and therefore provides quite realistic winds even at altitudes somewhat above 100km.

https://jawara.nipr.ac.jp/home

Sato, K., Tomikawa, Y., Kohma, M., Yasui, R., Koshin, D., Okui, H., et al. (2023), Interhemispheric Coupling Study by Observations and Modelling (ICSOM): Concept, Campaigns, and Initial Results, Journal of Geophysical Research Atmospheres, 128(11), e2022JD038249, doi:10.1029/2022JD038249.

Koshin, D., Sato, K., Watanabe, S., & Miyazaki, K. (2025), The JAGUAR-DAS whole neutral atmosphere reanalysis: JAWARA, Progress in Earth and Planetary Science, 12:1, https://doi.org/10.1186/s40645-024-00674-3.

Thank you for the recommendation. We will consider this as a possible future project.

(2) Is there a reason why you are not comparing to TIDI winds?

TIDI has a much longer dataset than MIGHTI.

We compared with MIGHTI instead of TIDI for a few reasons. The main reason, in this paper, is that we are comparing ACE v.5.3 winds with MIGHTI to show the improvement from v.5.2, which were previously compared by Johnson et al. (2024). The reason Johnson et al. chose MIGHTI was due to a higher number of coincidences with ACE measurements in the given time range compared with TIDI. This was also noted in Boone et al. (2021), but there is potential to compare with TIDI in the future.

(3) 1.59-66: You should specify whether data from a free WACCM run were used in your study, or whether WACCM was at least nudged to analysis/reanalysis at low altitudes. This information

is important because, depending on setup, WACCM data may be closer or less close to the real atmospheric state.

L297 in the preprint, now L310, states SD WACCM-X 2.2 was used. Added a sentence to clarify that this model is altered by observations. It now reads as "Specified Dynamics (SD) WACCM-X Version 2.2 provides global vector winds in 3 hour intervals. This SD version of WACCM-X uses observational data to produce wind speeds closer to the actual atmospheric state."

(4) In L.119 you mention that after homogenization ACE altitude profiles are shifted as a whole to match an analysis of the Canadian weather model at Environment and Climate Change Canada.

Please provide some information how large these shifts typically are.

The magnitude of the wind calibration shift was not tracked, so we cannot provide a typical value; however, we added an explanation. Now reads as: "As with previous processing versions, the final wind profile is shifted such that the results between 18 and 24 km match the expectations from an analysis run of the Canadian weather model at Environment and Climate Change Canada (Buehner, 2015). This is needed to account for the motion of the satellite relative to the atmosphere. A rough calibration of the ACE-FTS wavenumber scale is accomplished using high altitude CO\_2 lines, but because this process uses sampled peaks (which may not be sampled at line centers), the calibration could be off by a fraction of the width of the instrumental line shape 0.02 cm-1. The accuracy of this calibration will vary from occultation to occultation, but the wind calibration shifts can be as high as a few hundred m/s to compensate for the resulting offsets."

(5) 1.203: What do you think is the reason for the mentioned outliers?

Added sentence to explain. Now reads as: "Data Set 2 used only the coincidences where the average difference between ACE and MIGHTI wind speeds was less than 60 m/s, leaving us with 184 sunrises and 196 sunsets. The outliers removed are typically associated with a limited altitude window for the MIGHTI measurement."

(6) 1.207: Where do you think the sunrise/sunset biases come from?

Could this be some thermal drift of the satellite, or an effect of stray light that would be different between sunrise/sunset?

We are unsure about the cause of the sunrise-sunset at this time. It is something we plan on trying to solve in future versions.

(7) Another bias becomes evident in Fig.6b. At 100km ACE sunset winds are offset by 40m/s with respect to the radars. As measurements are coincident, this should not be an effect of atmospheric tides.

Do you have any idea where this offset comes from?

Added clarity so it now reads as: "Unlike the comparison with MIGHTI, the number of coincidences here is small, so the entirety of the coincident data set is considered. This means, unlike in MIGHTI where we removed outliers created by measurements with small altitude windows, outliers persist towards the edges of the meteor radar altitudes. This is best seen at 100 km in Fig. 6b, where the difference in Meteor Radar versus ACE is nearly 40 m/s at the top of the window."

(8) In Fig.7a ACE v5.3 shows a strong 50m/s jump at 50km, not seen in V5.2, or MERRA2. How often do such effects occur? Do you have any explanation for this effect?

This is already addressed in the text: "Near 52 km in Fig. 7(a), there is a sharp eastward increase in ACE v.5.3 wind speed. This shift occurs at the boundary of segments 2 and 3. There was probably an issue at the top of the segment 2 retrieval or at the bottom of the segment 3 retrieval for this occultation."

For clarity, rewriting as: "Near 52 km in Fig. 7(a), there is a sharp eastward increase in ACE v.5.3 wind speed. This shift occurs at the boundary of segments 2 and 3 and is likely an issue with the retrieval at the top of segment 2 or bottom of segment 3 (see Fig. 1) for this particular occultation."

**(9) About Fig. 10a:**

It is quite encouraging how well ACE captures the general global circulation patterns!

You should also mention that in September/October the winds in the tropics at 50km (eastward) and 80km (westward) are opposite. This is as expected from the vertical structure of the semiannual oscillation (SAO). See, for example, Ern et al. (2021), their Figs. 2 and 3.

It is also notable that the westward winds at 80km in the tropics are much stronger than in

MERRA2. From Ern et al. (2021), Fig.2 it looks like MERRA2 winds are strongly damped above 65km.

Added note and citation: "ACE measures the polar vortex near 30 km altitude in September-October. As expected, the vortex is weak in March-April. In September-October, the mesosphere is characterized by strong positive zonal winds in the whole hemisphere. It is well known that winds of the thermosphere are variable with local time. Because of ACE observation geometry, the sampled local times for sunrises (March-April) are typically between 6:00-am-and 9:00-am-and the sampled local times for sunsets (September-October) are typically between 15:00-pm and 18:00-pm. For sunrises in March-April, the thermosphere zonal winds are positive around 100 km altitude and negative above. At sunsets in September-October, strong negative zonal winds are observed at 60° S and altitude 130 km. Notably, the winds in the tropics are in opposite directions at 50 km (eastward) and 80 km (westward) in September-October. This is representative of the vertical structure of the semiannual oscillation (Ern et al, 2021)."

(10) Data availability section is missing.

Added section. Text reads as: "ACE v.5.2 and v.5.3 wind data can be found on the SCISAT webpage (https://databace.scisat.ca) within the Level 2 Data."

**TECHNICAL COMMENTS:**

1.26: atom oxygen -> atomic oxygen

Corrected.

1.228: able compare -> able to compare

Corrected.

---

## Author Comment (AC2)

**RC2 Comments**

**Specific Comments:**

In Figure 8 the MERRA-2 profile between 40 and 60 km jumps around 10 m/s between altitudes. This is surprising for averaged model data over 3000 profiles. Can the authors double check that these in fact are coming out of the MERRA-2 data and if real, add a statement to the paper explaining why this is occurring.

The jumpiness in the figure is "real" in that it is representative of the data, but in reality it is likely more smooth. Each MERRA-2 profile has slightly different heights which creates the variations described. Added clarity to the MERRA-2 section:

"All ACE line-of-sight wind speed measurements in 2019 and their corresponding MERRA-2 wind speeds are averaged and compared in Fig. 10. Note that there are ~10 m/s jumps in MERRA-2 wind speeds, best seen between 40 and 60 km. These are not real and are due to each MERRA-2 profile having data available at varying altitudes. However, the profile produced is real. There is a large profile disagreement between 40 and 60 km for ACE v.5.2 sunrises that now shows good agreement in v.5.3. Comparing the sunsets, we see similar profile agreement but a decrease in bias, particularly at lower altitudes."

Also added a note in the Fig 10 caption:

"(a) Average wind speed profiles from all ACE sunrise measurements in 2019 for ACE v.5.2 (3535 occultations), ACE v.5.3 (3535 occultations), and MERRA-2. (b) Average wind speed profiles from all ACE sunset measurements in 2019 for ACE v.5.2 (3419 occultations), ACE v.5.3 (3422 occultations), and MERRA-2. MERRA-2 wind speeds were converted using the ACE v.5.3 heading angles. Note that the sharp jumps in MERRA-2 wind speeds are a product of our processing of the MERRA-2 data, which have varying altitudes in each vertical profile."

Why does Figure 9 not have a legend like the rest of the Figures? The same strangeness in Figure 8 is seen in the differences as well.

Added legends to each tile of Figure 9.

On line 286 the paper states that "It is not reasonable for a model to predict wind patterns within the polar vortex during winter...". There are models that reasonably predict this wind pattern (MERRA-2 for example). This is just a reference to HMW14 and not models in general?

Changed to: "It is not reasonable for the climatology to predict wind patterns within the winter polar vortex, so measurements during those periods are removed."

It appears that in Figure 12, there is an error in plotting. Figures 12a-c are the same as Figures 12d-f. Please fix this and make sure the discussion of this figure in the body of the work is still correct with the correct figures.

Updated tiles d-f. The original text was based on the correct plot, so no changes were made.

---

## Author Comment (AC3)

**RC3 Comments**

**Specific Comments:**

1. The title could be made more informative by indicating the specific updates or changes that the paper addresses. As written, it is hard to guess the details of the study contained in the paper.

Changed to "Atmospheric Chemistry Experiment (ACE) v.5.3 Winds: Validation and Model Comparisons"

2. Overall, the introduction currently reads as a list of different datasets and models that describe winds across various atmospheric regions. As written, it is somewhat difficult for the reader to discern the main focus of the paper. Certain details—such as the spatial resolution of specific models—feel out of place for an introduction, while broader contextual framing is missing. The introduction should more clearly establish what this paper contributes and why it matters within the broader landscape of neutral wind research.

Made some large changes here. Removed "duplicate" examples to condense this section. Duplicate meaning different instruments that used similar techniques and/or covered the same region. The emphasis is that ACE provides wind data across all the ranges covered by the other instruments, and that it is still operating after 20 years. Also moved details unnecessary in the introduction about the models to their respective sections.

**The changes in the introduction now reads as:**

"Wind measurements by ACE are taken from the lower stratosphere through the lower thermosphere; however, most instruments only cover a small altitude range in comparison. Horizontal wind speeds are available in the troposphere and lower stratosphere through measurements from airplanes (Khelif et al., 1999) and balloons (Duruisseau et al., 2017; Kumer et al., 2014), ground-based lidar (Martner et al., 1993), and satellite lidar from the ADM-Aeolus (Atmospheric Dynamics Mission Aeolus) (Stoffelen et al., 2005; Lux et al., 2020). Ground-based lidar measurements have also been successful in the middle atmosphere (Baumgarten, 2010; Liu et al., 2002). Vector wind measurements in the upper mesosphere lower thermosphere (UMLT) can be recorded from space using Doppler shifts in airglow lines such as from atomic oxygen. A current example of this is Examples of this are the TIMED Doppler Interferometer (TIDI) instrument on the Thermosphere Ionosphere Mesosphere Energetics and Dynamics (TIMED) satellite (Killeen et al., 2006). Previously, the and previously from the Michelson Interferometer for Global-High-resolution Thermospheric Imaging (MIGHTI) instrument on the Ionospheric Connection Explorer (ICON) satellite used a similar technique. (Englert et al., 2017). and the Wind Imaging Interferometer (WINDII) on the Upper Atmosphere Research Satellite (UARS)

\text{\citep{shepherd1993}. Also on UARS, On Upper Atmosphere Research Satellite (UARS), the High Resolution Doppler Imager (HRDI) used O2 emission to measure winds in the UMLT (Hays et al., 1994; Grassl et al., 1995). Ground-based meteor radar (Liu et al., 2002; Tang et al., 2021) can also provide winds in the mesosphere. Line-of-sight winds near the mesopause have also been derived from the Doppler shift in O2 emission lines by the Microwave Limb Sounder (MLS) instrument on the Aura satellite (Wu et al., 2008). Note that, since many of these missions are inactive, ACE winds are especially valuable in this region. Since many of the missions mentioned are inactive, only measure a portion of the altitudes ACE covers, or only cover a small portion of the globe, the line-of-sight winds from ACE are especially valuable.

There have previously been technical issues preventing wind measurements in the middle atmosphere (around 30\$~\$\unit{km}\$ to 70\$~\$\unit{km}\$) \citep{baumgarten2010}, rufenacht2018}. There has been some success with ground-based lidar \citep{baumgarten2010}, liu2002} and microwave Doppler wind radiometers \citep{kumar2015}, but only up to the lower stratosphere. Sounding rockets \citep{schmidlin1985} can be used but are expensive and do not provide continuous measurements over long periods of time. Ground-based meteor radar \citep{liu2002, tang2021} can also provide winds in the mesosphere. There have been successful space-based missions that have provided line-of-sight winds in the middle atmosphere, such as the Atmospheric Trace Molecule Spectroscopy (ATMOS) experiment flown on the Spacelab 3 shuttle \citep{vanCleef1987} and the Superconducting Submillimeter-Wave Limb Emission Sounder (SMILES) mission \citep{baron2013} on the International Space Station.}

In this work, three wind datasets are directly compared to the new v.5.3 of ACE winds. The Modern-Era Retrospective Analysis for Research and Applications, version 2 (MERRA-2) produced by NASA's Global Modeling and Assimilation Office (GMAO) is an atmospheric reanalysis model based on modern satellite observations. MERRA-2 provides various data collections that contain information about many climate indicators, including atmospheric wind speeds. An in depth in-depth explanation of the model is available from Gelaro et al. (2017). MERRA-2 uses the Goddard Earth Observing System (GEOS) atmospheric model (Rienecker et al., 2008; Molod et al., 2015) and the Gridpoint Statistical Interpolation (GSI) analysis scheme (Kleist et al., 2009). The model uses a cubed-sphere horizontal discretization giving an approximate resolution of \$0.5\degree \times 0.625\degree\$ and 72 hybrid-eta (altitude) levels \cdot \cdot \times \left\{\text{putman2007}\}. These levels reach from the surface up to \$0.01\$\$-\$\underline{\text{unit}{hPa}} (around 75\$-\$\underline{\text{unit}{km}}) \citep{gelaro2017}, The model reaches to near 75 km, overlapping with which overlaps the lower half of ACE data.

Horizontal Wind Model Version: 2014 (HWM14) is an empirical climatology model of horizontal winds ranging from the troposphere up through the thermosphere. A detailed description of the climatology is available from Drob et al. (2015). The model began as HWM87 \citep{hedin1988} and has had multiple iterations since, each time adding more data from new instruments \citep{drob2015, hedin1996, drob2008}. The newest version, HWM14, HWM14 uses 73 x 106

observation measurements from 44 different instruments and a set of spherical harmonics to provide a statistical view of vector winds ranging from near the surface up to 1000 km. Because HWM14 is an empirical climatology, vector winds can be found for any given latitude and longitude.

The Whole Atmosphere Community Climate Model with thermosphere and ionosphere eXtension (WACCM-X) is produced by the National Center for Atmospheric Research (NCAR). WACCM-X is a comprehensive numerical model of the whole atmosphere, ranging from Earth's surface up to around 700 km (Liu et al., 2018). The model is an altitude extension of WACCM6 (Gettelman et al., 2019), which reaches up to ~140 km. WACCM outputs many climate and weather data products and is unique in that the model can be coupled with others to include ocean, sea ice, and land components.

Using CESM2 as a framework, WACCM is a mesh of NCAR projects: High Altitude Observatory (HAO) in the upper atmosphere, Atmospheric Chemistry Observations & Modeling (ACOM) in the middle atmosphere, and Climate & Global Dynamics (CGD) in the lower atmosphere. WACCM-X provides global vector winds on a \$1.9\degree \times 2.5\degree\$ grid."

**The changed MERRA-2 section now reads as:**

"MERRA-2 provides instantaneous 3-dimensional 3-hourly horizontal vector winds. The model uses cubed-sphere horizontal discretizations, giving an approximate resolution of  $0.5^{\circ}$  x  $0.625^{\circ}$  and 72 hybrid-eta (altitude) levels (Putman and Lin, 2007). The model is that are best constrained up to  $\sim$  40 km, but extends up to  $\sim$  70 km (Putman and Lin, 2007). Because the model is a reanalysis, we ..."

**The changed WACCM-X section now reads as:**

"Specified Dynamics WACCM-X Version 2.2 provides global vector winds in 3 hour intervals on a  $1.9 \circ \times 2.5 \circ 300$  grid. This SD version of WACCM-X uses observational data to produce wind speeds closer to the actual atmospheric state. WACCM-X vector winds were compared with ACE-FTS line-of-sight winds for all of 2019 in a similar fashion as HWM14...."

3. In the discussion of Figure 2, a brief description of the satellite's orbit and precession would also help orient readers attempting to interpret the figure.

Added clarity to this in the introduction of the paper. Now reads as: "The Atmospheric Chemistry Experiment (ACE) mission on board the Canadian satellite SCISAT is used for remote sensing Earth's atmosphere (Bernath et al., 2005; Bernath, 2017). The SCISAT satellite operates in a low Earth near polar orbit with an inclination of 73.9°. The ACE mission uses limb geometry ..."

4. Lines 200–201: The terms "similar agreement" and "better agreement" should be supported with quantitative similarity metrics rather than qualitative, visual judgments. This comment applies throughout the paper wherever such comparisons are discussed. Since a quantitative comparison was performed for the MERRA-2 results, a comparable level of quantitative analysis would also be expected for the MIGHTI, meteor radar, and other model comparisons.

Added figures similar to the MERRA-2 bias plot (Fig.9 in preprint) for the other models and instruments. Have also adjusted / added text accordingly.

**Changes to MIGHTI text:**

"The sunset comparison shows good agreement for both versions from 90 to 110\$~\$\unit{km} and then deviates. There is a sunrise (sunset) bias of around \$+15\$\$~\$\unit{m/s}} (\$-15\$\$~\$\unit{m/s}), shown in pink in Fig. \ref{fig:MIGHTIdataset2}.

We are also able to derive a sunrise-sunset bias with this comparison. To do this, we shift the MIGHTI altitude to the nearest ACE altitude (maximum of  $\pm 0.5$  km) and find the difference between ACE and MIGHTI wind speeds. The differences are then averaged at each altitude over all occultations in Data Set 2 and the results are shown in Fig. 5(a) for v.5.2 and (c) for v.5.3. We then subtract the sunrise and sunset averages from the total average to find the bias. The average bias at each altitude is displayed in Fig. 5(b) for v.5.2 and (d) for v.5.3. We find that the sunrise (sunset) bias for v.5.2 is generally within  $\pm 5$  m/s of the previously found -15 m/s (+15 m/s) up to 120 km. The bias for v.5.3 is within  $\pm 2$  m/s of the -15 m/s (+15 m/s) bias up through the same altitude. The sunrise-sunset bias is shown in pink in Fig. 4."

**Changes to Meteor Radar text:**

For the sunrises, we again see well matching profiles. The sunsets we see general agreement but with ACE showing more prominent features. The sunrise (sunset) bias of about \$+15\$\$~\$\unit{m/s} (\$-15\$\$~\$\unit{m/s}) is evident in this comparison.

[revised manuscript text omitted]

5. The ±15 m/s adjustment described is somewhat unclear. In Figure 4, it appears that 15 m/s has been added to the sunrise case and subtracted from the sunset case to produce the pink curve. Unlike the individual example in Figure 3, the average agreement between MIGHTI and ACE appears to worsen at sunrise between versions 5.2 and 5.3; if so, this should be explicitly acknowledged and, if possible, explained. The authors should also specify how the ±15 m/s offset was determined—was it chosen by eye or by optimizing an objective measure of agreement? The latter approach would strengthen the analysis and make it more reproducible. Moreover, rather than saying "There is a sunrise—sunset bias of around ±15 m/s," the phrasing should clarify the directionality, e.g., "At sunrise (sunset), there is a bias of approximately –15 m/s (+15 m/s)." This clarification should be applied in the Figure 4 caption, the abstract, and anywhere else this statement appears.

Changed to suggested format: "sunrise (sunset)" and "+15 m/s (-15 m/s)" throughout the paper. In context where we are discussing the "sunrise-sunset" bias without specific values, I have left

it as is. As for how the bias was determined, refer to comment 4 as we have further quantified each model / dataset similar to what was done with MERRA-2.

**Technical Corrections:**

Line 26: Typo. Should read "atomic oxygen"

Corrected.

Line 64: The acronym CESM2 is used without being defined earlier in the text.

Now reads as: "... Using CESM2 (Community Earth System Model 2) as ..."

In Figure 2, the x-axis should be limited to 0–360 rather than extending to 400 to avoid confusion.

*Updated x-axis to 0-365 days and the y-axis to 0-360 degrees, rather than 0-400 for both.*

In the description of the MIGHTI data, the authors should additionally cite Englert et al. (2023), which describes the v05 MIGHTI wind product used in the comparisons.

Citation now includes Englert et al. (2017) and (2023).

Line 190: specify that the winds retrieved are horizontal vector winds, not the full 3D wind vector.

Added "horizontal" for clarity: "These measurements can be used to determine horizontal vector winds due to ..."